



# Uncertainties in mass balance estimation of the Antarctic Ice Sheet using the input-output method

Yijing Lin[1,2*], Yan Liu[1,3*], Zhitong Yu[2]†, Xiao Cheng[1, 3, 4], Qiang Shen[5], Liyun Zhao[1, 3]

[1]College of Global Change and Earth System Science (GCESS), Beijing Normal University, Beijing 100875, China & South
ern Marine Science and Engineering Guangdong Laboratory (Zhuhai), Zhuhai 519082, China;

[2]Qian Xuesen Laboratory of Space Technology, China Academy of Space Technology, Beijing 100094, China;

[3] Southern Marine Science and Engineering Guangdong Laboratory, Zhuhai 519082, China

[4] School of Geospatial Engineering and Science, Sun Yat-Sen University, Zhuhai 519082, China

[5]State Key Laboratory of Geodesy and Earth's Dynamics, Innovation Academy for Precision Measurement Science and
Technology, CAS, Wuhan 430077, China;

*These authors contributed equally to this work.

† Correspondence to: Z. Yu (yuzhitong@qxslab.cn)

**Abstract.** The input-output method (IOM) is one of the most popular methods of estimating the ice sheet mass balance (MB), with a significant advantage in presenting the dynamics response of ice to climate change. Assessing the uncertainties of the MB estimation using the IOM is crucial to gaining a clear understanding of the Antarctic ice-sheet mass budget. Here, we introduce a framework for assessing the uncertainties in the MB estimation due to the methodological differences in the IOM, the impact of the parameterization and scale effect on the modeled surface mass balance (SMB, input), and the impact of the uncertainties of ice thickness, ice velocity, and grounding line data on ice discharge (D, output). For the uncertainty assessment of the D, we present the D at a fine scale. In setting a goal of determining the Antarctic MB within an uncertainty of 15 Gt yr$^{-1}$, we found that the different strategies employed in the methods cause considerable uncertainties in the annual MB estimation. The uncertainty of the RACMO2.3 SMB caused by its parameterization can reach 20.4 Gt yr$^{-1}$, while that due to the scale effect is up to 216.7 Gt yr$^{-1}$. The observation precisions of the MEaSUREs InSAR-based velocity (1−17 m yr$^{-1}$), the airborne radio-echo sounder thickness ($\pm$100 m), and the MEaSUREs InSAR-based grounding line ($\pm$100 m) contribute uncertainties of 17.1 Gt yr$^{-1}$, 10.5$\pm$2.7 Gt yr$^{-1}$ and 8.0~27.8 Gt yr$^{-1}$ to the D, respectively. However, the D uncertainty due to the remarkable ice thickness data gap, which is represented by the thickness difference between the BEDMAP2 and the BedMachine reaches 101.7 Gt yr$^{-1}$, which indicates that it is the dominant cause of the future D uncertainty. In addition, the interannual variability of the D caused by the annual changes in the ice velocity and ice thickness are considerable compared with the target uncertainty of 15 Gt yr$^{-1}$, which cannot be ignored in annual MB estimations.





**1 Introduction**

The Antarctic ice sheet (AIS) contains 88% of the global land ice, with a sea level equivalent of 58.3 m (Bamber et al., 2018), making it the largest potential source of future sea level rise and the most uncertain contributor to future sea level projections (Oppenheimer et al., 2019). As an important indicator of global climate change, even minor changes in the Antarctic ice volume have significant impacts not only on the global mean sea level but also on the hydrological cycle, carbon cycle,

atmospheric circulation, sea surface temperature, sea salinity, and thermohaline circulation (Thomas et al., 2017; Wadham et al., 2019; Smith et al., 2020). For decades, estimating the Antarctic mass balance (MB) and understanding how, where, and why the ice sheets lose mass have been the main goals of Antarctic research (Kennicutt et al., 2014). The need for a more accurate determination of the Antarctic mass budget has become imperative for gaining a better understanding of its dynamics, for improving our understanding of ice sheet evolution, and for enabling more accurate future projections of the global sea

level (Shen et al., 2018). However, there is still a large uncertainty in the assessment of the ice sheet MB, and there are significant differences among the most recent results obtained using different methods due to the limitations of the methods used and the lack of data sources (Shepherd et al., 2018; Rignot et al., 2019; Smith et al., 2020; Hugonnet et al., 2021; Wang et al., 2021). A detailed understanding of the uncertainty sources is conducive to accurately estimating the ice sheet MB and to decreasing the uncertainties in the ice sheet's contribution to sea-level rise and determining its influencing mechanism

(Massom and Lubin, 2006).

The four main methods for estimating the MB of ice sheets, based on different types of satellite observations include satellite altimetry (Helm et al., 2014; McMillan et al., 2014; Zwally et al., 2015; Smith et al., 2020), gravimetry (Velicogna et al., 2014; Forsberg et al., 2017), the input-output method (IOM) (Rignot et al., 2008; Rignot et al., 2011a; Rignot et al., 2019), and the reconciled estimate (Shepherd et al., 2012; Shepherd et al., 2018). As a unique method that can determine the physical

processes responsible for mass loss, the IOM is the most advantageous for understanding the mass changes in each basin. It quantifies the glacier mass gained through snowfall and the loss via sublimation, meltwater runoff, and ice discharge into the ocean, allowing us to separately examine the changes in the surface mass balance (SMB) and ice dynamics on the scale of individual glacier drainage basins.

However, the IOM involves numerous parameters and complicated calculations, whose uncertainties result in large

differences in the mass balance estimates (Bamber et al., 2018). In addition to the large uncertainty of up to three times those of altimetry and gravimetry, IOM estimates have a greater bias from the average result, which requires a more detailed analysis of the source of the differences to clarify whether this bias is significant or systematic (Shepherd et al., 2018). In particular, while the West Antarctic Ice Sheet (WAIS) and the Antarctic Peninsula Ice Sheet (APIS) are consistently thought to be in a state of continuous mass loss, whether the East Antarctic Ice Sheet (EAIS) is losing or gaining mass is still controversial. For

the EAIS, in contrast to the mass gain of $58\pm43$ Gt yr$^{-1}$ from 2003–2020 obtained using the gravity method (Wang et al.,





2021) and $90\pm21$ Gt yr$^{-1}$ from 2003–2019 obtained using the altimetry method (Smith et al., 2020), mass losses of $82\pm9$ Gt yr$^{-1}$ from 1999–2009 and $51\pm13$ Gt yr$^{-1}$ from 2009–2017 were estimated by Rignot et al. (2019) using the IOM.

In addition to the conflicting results obtained using the IOM and other methods, the estimates of studies based on the IOM still have significant differences due to data differences and methodological differences. The Antarctic mass balance

estimate for 2008 obtained by Shen et al. (2018) has a maximum difference of 34.0 Gt yr$^{-1}$ compared that obtained by Gardner et al. (2018), and 35.1 Gt yr$^{-1}$ compared to that obtained by Shepherd et al. (2018). Similarly, there are significant differences in the estimates based on the IOM for typical drainage basins. For example, Rignot et al. (2019), Wen et al. (2007), and Zhou et al. (2019) all estimated the MB of the Lambert (basin C-D), with a maximum difference of 5.9 Gt yr$^{-1}$. For Pine Island, the MB estimates of Rignot et al. (2019) and Shen et al. (2018) have a maximum difference of 17.9 Gt yr$^{-1}$.

Although several studies have assessed and analyzed the data uncertainties of several parameters in detail to clarify the influence of the potential uncertainty in the IOM (Rignot et al., 2011c; Shepherd et al., 2012; Bamber et al., 2018; Gardner et al., 2018; Shen et al., 2018; Rignot et al., 2019), the dominant factors influencing the uncertainties in the MB estimation of the AIS remain unclear. For example, Rignot et al. (2019) compared the SMBs derived from the regional climate models RACMO2.3 version p1 and version p2 in selected basins. Shepherd et al. (2012) assessed the differences between the SMB of

RACMO2 and in situ accumulation observations. In their second ice sheet mass balance inter-comparison exercise (IMBIE) assessment, Shepherd et al. (2018) compared the SMB products from different regional climate models and re-analysis models. These studies mainly focused on the data uncertainty of the SMB but ignored uncertainties associated with the other parameters. In addition, subtle differences in the methods used could also introduce uncertainties. The IOM is still being refined (Rignot and Thomas, 2002), so the exact calculation process varies from study to study, which has often been overlooked. The majority

of previous studies have focused on the consistency between the MB estimates obtained using different methods (Rignot et al., 2011c; Shepherd et al., 2012; Shepherd et al., 2018) but have ignored the potential impact of methodological differences on the mass balance estimates.

In this study, we introduce an uncertainty analysis framework for comprehensively and systematically assessing the methodological uncertainty and data uncertainty in the MB estimation of the AIS using the IOM. By combining multi-source

satellite datasets, first we quantify the uncertainties of the Antarctic MB estimate due to methodological differences, different datasets for each parameter, and data errors. Our framework enables the determination of the predominant uncertainty source and the underlying causes, as well as comparison among multiple datasets and different methods. Section 2 describes the methodology employed in this study, including the IOM and uncertainty analysis, and the data used. Section 3 presents the results corresponding to each scheme. The results are discussed in Section 4, and our conclusions are presented in Section 5.



## 2 Data and Methodology

The uncertainty in the AIS MB estimation using the IOM is divided into methodological uncertainty and data uncertainty. The methodological uncertainty arises from the methodological differences in the assessment of the MB components. The data uncertainty is mainly determined by the different datasets used and the measurement errors (i.e., measurement accuracy) due to systematic errors and random errors (i.e., measurement precision).

### 2.1 Input-output method

The IOM quantifies two individual components of the ice sheet MB in a year, i.e., the SMB (input) and ice discharge (D, output). The former, as the input component, is comprised of precipitation (snowfall or rainfall) minus surface ablation. In the absence of significant melt and when the snow sublimation and erosion are an order of magnitude smaller than the snowfall, the SMB of the ice sheet is mostly determined by the snowfall (Lenaerts et al., 2014). Because of the lack of temporally and spatially continuous in situ observations (Agosta et al., 2019), the SMB is often modeled and estimated using regional climate models (RCMs). In the calculation of the SMB, glaciers are commonly lumped into larger units for simplification, such as regions, basins, and interior sub-basins, and thus, the accuracy and consistency of the drainage boundary are crucial to the estimation.

D is the grounding line ice flux that is calculated as the flux gate width multiplied by the ice velocity and ice thickness. Previous estimates of D were usually presented in units of region or drainage basin, making it difficult to distinguish the dominant cause of the uncertainties associated with D, and to determine the effects of each parameter on D. Because the ice velocity and ice thickness vary considerably at the pixel scale, we estimated D at a fine scale by discretizing the grounding line into grids of the same cell size (1 km) as the ice thickness and ice velocity data, which divided the MEaSUREs InSAR-based grounding line into 58,597 flux gates. $\vec{D}_i$ at the pixel scale is calculated as follows:

$$\vec{D}_i = H_i \cdot \vec{V}_i \cdot \vec{L}_i \cdot \rho_{ice},\tag{1}$$

where $H_i$ is the equivalent ice thickness of flux gate $i$, $\vec{V}_i$ is the ice velocity along the ice flow direction, and $\vec{L}_i$ is the fluxgate width across the ice flow direction.

### 2.2 Data and methods for uncertainty analysis

The IOM parameters include the SMB, and the D parameters of ice thickness, ice velocity, grounding line, and drainage boundary. To assess the data uncertainty of these components (D and SMB) and parameters, we selected the representative data products while considering the data availability, and their spatial and temporal coverages (Table 1).



**Table 1. Data catalogue**

| Dataset | Measuring Method | Temporal Coverage | Data format | Data Source |
|---|---|---|---|---|
| SMB (RACMO2.3p1) (van Wessem et al., 2014) | ERA-Interim re-analysis | 1979–2016 | AN: 35 km × 35 km raster<br>AN: 27 km × 27 km raster | KNMI |
| SMB (RACMO2.3p2) (van Wessem et al., 2016; van Wessem et al., 2018) | ERA-Interim re-analysis | 1979–2016 | AN: 27 km × 27 km raster<br>AP: 5.5 km × 5.5 km raster | KNMI |
| Thickness (Bedmap2) (Fretwell et al., 2013) | Airborne radar altimetry | 1970–2000 | 1 km × 1 km raster | BAS |
| Thickness (BedMachine V2) (Morlighem et al., 2020) | Airborne radar altimetry; Mass conservation method | 1970–2019 | 500 m × 500 m raster | NASA |
| Velocity (MEaSUREs) (Mouginot et al., 2012) | InSAR | 1996–2016 | 450 m × 450 m raster | NASA |
| Velocity (MEaSUREs) (Mouginot et al., 2019) | InSAR | 1996–2018 | 450 m × 450 m raster | NASA |
| Velocity (MEaSUREs) (Mouginot et al., 2017) | InSAR | 2005–2016 | 1 km × 1 km raster | NASA |
| Grounding line (MEaSUREs) (Rignot et al., 2013) | DInSAR | 1992–2014 | Polyline | NASA |
| Drainage boundary (MEaSUREs) (Rignot et al., 2013) | InSAR | 1992–2015 | Polygon | NASA |

We assessed the uncertainties in the MB estimation of the AIS in three separate ways: the D and MB uncertainty due to

the methodological differences; the SMB differences due to the parameterizations in different model versions and the scale effect of the products at different spatial scales; and the D uncertainty due to the uncertainties of the ice thickness, velocity, and grounding line data. The analysis framework and the data schemes used in the uncertainty assessment are presented in Table 2.

In the assessment of the methodological uncertainty in the MB, first, we assessed the possible differences between the D

estimated using full velocity ranges and the D estimated from the high velocity region using a scaling factor. In the last four decades of MB estimations (Rignot et al., 2019), the scaling factor which was calculated over and validated on the fastest parts of the glacier, was applied to the slow-moving glacier to obtain a complete time series of the yearly D. This approach effectively fills the data gap but ignores the response of low-velocity glaciers to climate change, and it may have an unknown impact on the MB. We obtained the interannual variations in D for different ice velocity ranges by classifying D into three categories: D1

($V < 20$ m yr$^{-1}$), D2 (20 m yr$^{-1}$ $< V <$ 100 m yr$^{-1}$), and D3 ($V > 100$ m yr$^{-1}$). Then, we compared the D of the AIS obtained using the full velocity ranges with the D3 obtained using the scaling factor. In this study, the scaling D was estimated by applying the scaling factor of the fast flows with velocities greater than 100 m yr$^{-1}$ relative to their average velocity to the average D. Second, we assessed the impact of the SMB data strategy on the MB. The interannual change in the SMB for all AIS sectors is highly variable over a range of time scales. Previous studies have used different time regression methods to

constrain the variability of the SMB, such as using the multi-year average SMB instead of the annual SMB for the interannual MB estimation (Shen et al., 2018). This may characterize the secular trends in the AIS mass changes, but it omits the significant temporal variability. We compared the interannual MBs obtained using the multi-year average SMB and the annual SMB for the time period 2005–2016.





In the assessment of the SMB uncertainty due to the parameterization and scale effect, we compared the SMBs from two

versions of RACMO2.3, p1 and p2 (van Wessem et al., 2014; van Wessem et al., 2018), at three different resolutions (35 km,

27 km, and 5.5 km). The SMB data were averaged over the period 1979–2008, except for RACMO2.3p1 (35 km), which

corresponds to the 1979–2011 average. Among these, RACMO 2.3p2 (5.5 km) has the highest resolution, but only covers the

Antarctic Peninsula (AP). We used the MEaSUREs Antarctic boundary dataset for the estimates in different regions and basins

because this AIS boundary is consistent with the MEaSUREs grounding line.

We assessed the D response to the uncertainties of the ice thickness, velocity, and grounding line data. We used two ice

thickness products, the BEDMAP2 and the BedMachine, both of which cover the AIS. The BEDMAP2 was presented in 2013

(Fretwell et al., 2013) and the BedMachine in 2020 as the latest Antarctic thickness product (Morlighem et al., 2020). For the

assessment of the ice velocity uncertainty, we used two ice velocity datasets, the MEaSUREs InSAR-based velocity (Rignot

et al., 2011a; Mouginot et al., 2012) and the MEaSUREs Phase-based velocity (Mouginot et al., 2019). Both provide

comprehensive ice velocity maps over the past two decades and were assembled from the same multiple satellite data to cover

the entire AIS. The former was generated by speckle tracking (Rignot et al., 2011a; Mouginot et al., 2012), and the latter

combines interferometric phases and reaches centimeter-level precision over 80% of the AIS (Mouginot et al., 2019).

Furthermore, we assessed the D uncertainty due to errors in the ice thickness and ice velocity data. We estimated the

response of the D uncertainties to their system error and random error based on the reference D calculated using the reference

velocity of the MEaSUREs Phase-based velocity and the reference thickness of the BedMachine thickness. To estimate the D

response to random error, we randomly generated errors in the ice thickness and ice velocity for each flux gate. The generated

errors were normally distributed. Based on the precision of the data products used, we determined that the maximum random

error ranges corresponding to the ice thickness and ice flow rate are ±100 m and ±20 m yr$^{-1}$, respectively. We made corrections

for the generated random errors to avoid negative data values after introducing errors, as follows:

$$H_i' = \begin{cases} H_i + u_{H_i}, & H_i' \geq 0 \\ 1, & H_i' < 0 \end{cases} \tag{2}$$

$$V_i' = \begin{cases} V_i + u_{V_i}, & V_i' \geq 0 \\ 0.1, & V_i' < 0 \end{cases} \tag{3}$$

where $H_i$ and $V_i$ are the original ice thickness (m) and velocity (m yr$^{-1}$) of flux gate $i$, respectively, $u_{H_i}$ is the generated

random error of the thickness, $u_{V_i}$ is the generated random error of the velocity, $H_i'$ is the corrected ice thickness, and $V_i'$ is

the corrected ice velocity. The histograms of the random errors are shown in Fig. 1.



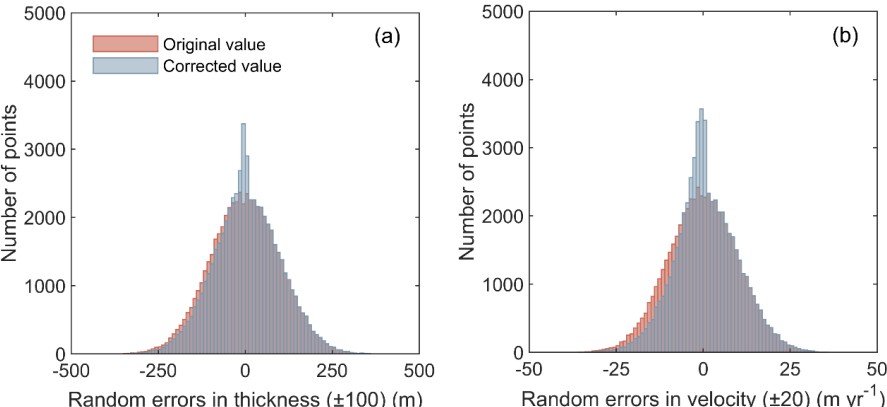

**Figure 1.** Histograms of the random errors in the (a) ice thickness, and (b) velocity. The orange bars are the initial normally distributed random errors and the blue bars denote the random errors corrected for negative numbers.

Finally, to investigate the D uncertainty due to the effect of the grounding line position on the D results, we generated three flux gates: the original grounding-line flux gate, the flux gate for a 1 km advance, and the flux gate for a 1 km retreat. To cover the frontal edge of the advanced ice sheet, we extended the area of the original raster dataset using the Moore-neighbor algorithm. The ice thickness and ice velocity values corresponding to the three flux gates were used to calculate their D value.

**Table 2. Experimental schemes for uncertainty analysis of the IOM**

| Component | Parameter | Data combination |
|---|---|---|
| **Methodological uncertainty** | | |
| MB (2005-2016) | MB ($SMB_{Annual}$) | Annual SMB (RACMO2.3p2, 27 km) + $D_{Full-range}$ + $T_{BedMachine}$ + GL |
| | MB ($SMB_{Average}$) | Multi-year average SMB (RACMO2.3p2, 27 km,) + $D_{Full-range}$ + $T_{BedMachine}$ + GL |
| D (2005-2016) | $D_{Full-range}$ | V(MEaSUREs Annual Velocity, 2005−2016) + $T_{BedMachine}$ + GL |
| | $D_{scaled}$ | V(MEaSUREs Annual Velocity, 2005−2016, applied scaling factors) + $T_{BedMachine}$ + GL |
| **Data uncertainty** | | |
| SMB | $SMB_{p1-35}$ | RACMO2.3p1, AN: 35 km |
| | $SMB_{p1-27}$ | RACMO2.3p1, AN: 27 km |
| | $SMB_{p2-27}$ | RACMO2.3p2, AN: 27 km |
| | $SMB_{p2-5.5}$ | RACMO2.3p2, AP: 5.5 km |
| D | **Velocity** | |
| | D ($V_{InSAR-based}$) | V(MEaSUREs InSAR-based) +T(BedMachine) +GL |
| | D ($V_{Phase-based}$) | V(MEaSUREs Phase-based) +T(BedMachine) +GL |
| | D ($V_{Phase-based}+e_{system}$) | V(MEaSUREs Phase-based+$e_{random}$) +T(BedMachine) +GL |
| | D ($V_{Phase-based}+e_{random}$) | V(MEaSUREs Phase-based+$e_{system}$) +T(BedMachine) +GL |
| | **Thickness** | |
| | D ($T_{Bedmap2}$) | V(MEaSUREs Phase-based) +T(Bedmap2) +GL |
| | D ($T_{BedMachine}$) | V(MEaSUREs Phase-based) +T(BedMachine) +GL |
| | D ($T_{BedMachine}+e_{system}$) | V(MEaSUREs Phase-based) +T(BedMachine+$e_{random}$) +GL |
| | D ($T_{BedMachine}+e_{random}$) | V(MEaSUREs Phase-based) +T(BedMachine+$e_{system}$) +GL |
| | **Grounding line** | |
| | D ($GL_0$) | V(MEaSUREs Phase-based) +T(BedMachine) +GL |





| D (GL$_{Advance}$) | V(MEaSUREs Phase-based) +T(BedMachine) +GL(Advance 1 km) |
| D (GL$_{Retreat}$) | V(MEaSUREs Phase-based) +T(BedMachine) +GL(Retreat 1 km) |

## 3 Results

### 3.1 Methodological uncertainty

The average annual D values of the EAIS, WAIS, APIS, islands, and the entire AIS for the time period 2005–2016 are

889.7 Gt yr$^{-1}$, 747.5 Gt yr$^{-1}$, 161.1 Gt yr$^{-1}$, 132.2 Gt yr$^{-1}$ and 1930.6 Gt yr$^{-1}$, respectively, with the standard deviations of

13.9 Gt yr$^{-1}$, 12.1 Gt yr$^{-1}$, 3.5 Gt yr$^{-1}$, 4.9 Gt yr$^{-1}$ and 21.7 Gt yr$^{-1}$, respectively. The average D for the entire AIS is 356.2 Gt

yr$^{-1}$ less than that estimated by Rignot et al. (2019) for the same observation period. D1 (V < 20 m yr$^{-1}$), D2 (20 < V < 100

m yr$^{-1}$), and D3 (V > 100 m yr$^{-1}$) account for 1.9%, 11.4%, and 86.7% of the average annual D value for the AIS,

respectively. The interannual variations in D for the different velocity ranges are obviously inconsistent, and the interannual

variation trends noted for D2 and D3 are opposite in some periods (Fig. 2a). However, since the D3 interannual variability is

significantly larger than those of the other two low velocity ranges (Fig. 2a), the interannual variability of D$_{scaled}$ estimated

from D3 is only 3.6 Gt yr$^{-1}$ greater than that of D$_{Full-range}$ (Fig. 2b). The largest difference between D$_{scaled}$ and D$_{Full-range}$ with a

value of 10.4 Gt yr$^{-1}$ occurred in 2006 when the corresponding D2 was the second largest and the corresponding D3 was the

smallest.

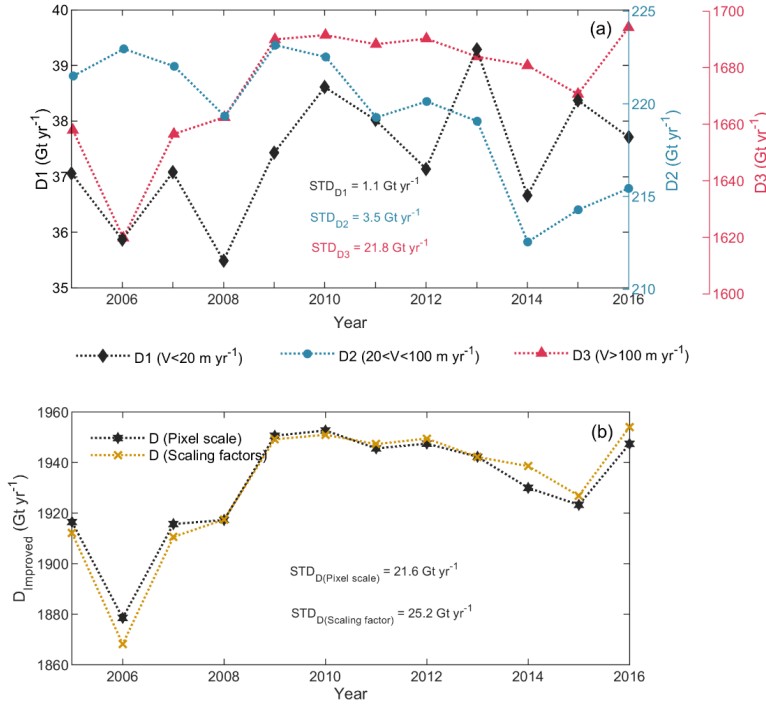



**Figure 2.** (a) Interannual variations in D for different velocity ranges over the period 2005–2016. D is represented by the lines of different colors and types corresponding to the axis colors, which are classified into three categories (V < 20 m yr$^{-1}$, 20 < V < 100 m yr$^{-1}$, and V > 100 m yr$^{-1}$). (b) Annual mass balance estimation of the Antarctic ice sheet from 2005 to 2016. The lines represent the MB estimation using the refined IOM and that using scaling factors derived from the high-velocity regions.

The average annual SMBs of the EAIS, WAIS, APIS, islands and the entire AIS for the period 2005–2016 are 1163.4 Gt yr$^{-1}$, 633.0 Gt yr$^{-1}$, 277.7 Gt yr$^{-1}$, 58.7 Gt yr$^{-1}$ and 2132.7 Gt yr$^{-1}$, respectively, with standard deviations of 96.6 Gt yr$^{-1}$, 66.0 Gt yr$^{-1}$, 42.6 Gt yr$^{-1}$, 4.5 Gt yr$^{-1}$, and 103.7 Gt yr$^{-1}$, respectively. In the EAIS, the SMB exhibited two significant positive anomalies in 2009 (178.0 Gt yr$^{-1}$) and 2011 (110.8 Gt yr$^{-1}$) compared with the 12-year mean, which were both caused by extreme snowfall episodes. In the WAIS, the SMB exhibited the largest positive anomaly in 2005 (142.8 Gt yr$^{-1}$) and the

largest negative anomaly in 2011 (−91.0 Gt yr$^{-1}$). The standard deviations of the SMBs are more than four times greater than the standard deviations of the D values for the EAIS, WAIS, and APIS regions, but the linear rate of decrease for the SMB (5.1 Gt yr$^{-1}$) is only two times greater than the linear rate of increase for D (2.6 Gt yr$^{-1}$). Fig. 3 shows the annual MBs for the EAIS and WAIS obtained using the annual SMB and the average annual SMB for the period 2005–2016. The standard deviations of the annual MBs (SMB$_{Annual}$) for the EAIS and WAIS are 93.1 Gt yr$^{-1}$ and 74.1 Gt yr$^{-1}$, respectively,

while those of the average annual MBs (SMB$_{Average}$) are 13.9 Gt yr$^{-1}$ and 12.1 Gt yr$^{-1}$, respectively. The average SMB strategy may provide the long-term trend of the MB, but it omits the significant short-term variability, which contributes to the remarkable disagreement between the estimates.

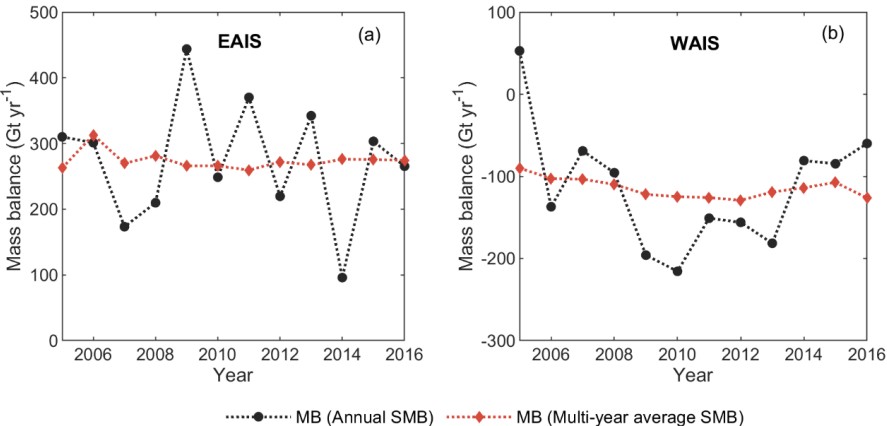

**Figure 3.** Comparison of the MBs obtained using the annual SMB and average annual SMB for the period 2005–2016.


## 3.2 Data uncertainty

### 3.2.1 SMB uncertainties

Fig. 4a shows the different SMB results from the various RCM versions and the different spatial resolutions. At the same spatial scale (27 km) but using parameterizations, the $SMB_{p1-27}$ is 20.4 Gt yr$^{-1}$ less than the $SMB_{p2-27}$ for the entire AIS. The differences between the $SMB_{p1-27}$ and $SMB_{p2-27}$ for the EAIS (78.5 Gt yr$^{-1}$) and the APIS (39.4 Gt yr$^{-1}$) are much greater than those for the islands (15.6 Gt yr$^{-1}$) and WAIS (3.1 Gt yr$^{-1}$) (Table 3). For the EAIS, the $SMB_{p1-27}$ and $SMB_{p2-27}$ in basins with relatively large SMBs (i.e., basin C-C' (23.1 Gt yr$^{-1}$), basin C'-D (15.4 Gt yr$^{-1}$), and basin D-D' (8.6 Gt yr$^{-1}$)) have relatively large differences. For the APIS, West Graham Land in basin I-I'' has a significant difference of 37.4 Gt yr$^{-1}$, accounting for 95% of the entire APIS difference.

**Table 3. Data uncertainties of the IOM (Gt yr$^{-1}$)**

| Components | EAIS | WAIS | APIS | Islands | AIS |
|---|---|---|---|---|---|
| **SMB** | | | | | |
| $SMB_{p1-35}$ | 1048.9 | 601.0 | 173.6 | 57.4 | 1880.9 |
| $SMB_{p1-27}$ | 1075.0 | 652.6 | 293.0 | 77.0 | 2097.6 |
| $SMB_{p2-27}$ | 1153.5 | 649.5 | 253.6 | 61.4 | 2118.0 |
| $SMB_{p2-5.5}$ | - | - | 305.3 | - | - |
| ABS(dif_max) | 104.6 | 51.6 | 131.7 | 19.6 | 237.1 |
| **D (Ice velocity)** | | | | | |
| $D(V_{InSAR-based})$ | 903.5 | 752.8 | 162.8 | 133.9 | 1953.1 |
| $D(V_{Phase-based})$ | 895.8 | 750.2 | 157.7 | 132.4 | 1936.0 |
| ABS(dif_max) | 7.7 | 2.6 | 5.1 | 1.5 | 17.1 |
| **D (Ice thickness)** | | | | | |
| $D(T_{BEDMAP2})$ | 914.2 | 783.1 | 183.4 | 157.1 | 2037.8 |
| $D(T_{Bedmachine})$ | 895.8 | 750.2 | 157.7 | 132.4 | 1936.0 |
| ABS(dif_max) | 18.4 | 32.9 | 25.7 | 24.6 | 101.7 |
| **D (Grounding line)** | | | | | |
| $D(GL_{Advance})$ | 798.6 | 643.1 | 113.3 | 102.9 | 1657.9 |
| $D(GL_0)$ | 895.8 | 750.2 | 157.7 | 132.4 | 1936.0 |
| $D(GL_{Retreat})$ | 887.0 | 742.7 | 156.9 | 69.9 | 1856.4 |
| ABS(dif_max) | 106.2 | 107.1 | 44.4 | 62.5 | 278.1 |

For the same version of RACMO2.3p1 but at different spatial scales, the $SMB_{p1-35}$ is 216.7 Gt yr$^{-1}$ smaller than the $SMB_{p1-27}$ for the entire AIS. The APIS accounts for 55.1% of the total difference, and West Graham Land in basin I-I'' accounts for 65.5% of the total difference in the APIS. Similarly, for version RACMO2.3p2 but at different spatial scales, the $SMB_{p2-27}$ is 51.7 Gt yr$^{-1}$ less than $SMB_{p2-5.5}$ for the APIS with a difference of 42.7 Gt yr$^{-1}$ in West Graham Land, accounting for 82.6% of the difference. Fig. 4b shows the distinct difference between $SMB_{p2-27}$ and $SMB_{p2-5.5}$ in West Graham Land over four decades, with an average of 51.4 Gt yr$^{-1}$. The difference has been gradually increasing since 2004, reaching the maximum value of 117.4 Gt yr$^{-1}$ in 2016.

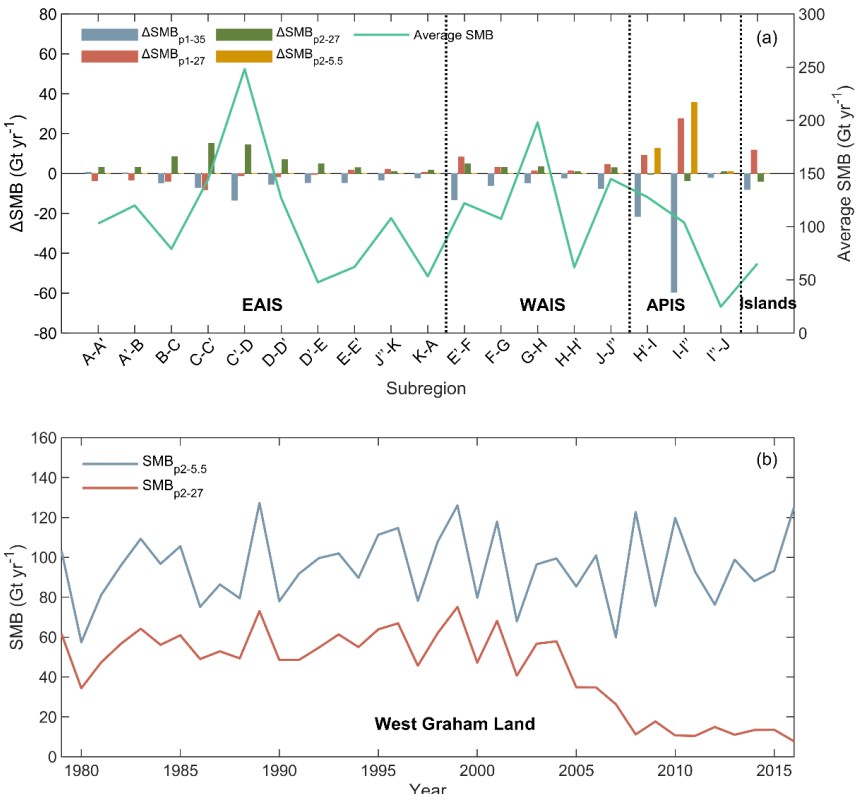

**Figure 4.** (a) The differences between the average SMB and the SMBs for the different RCM versions and different resolutions (colored bars, left axis). Also shown in (a) are the average SMB data (green line, right axis). (b) Comparison of the SMBs (RACMO2.3p2) at 5.5 km (blue line) and 27 km (red line) resolutions in West Graham Land, Antarctic Peninsula.

### 3.2.2 Discharge response to ice velocity uncertainty

Fig. 5 shows the uncertainty in D due to the systematic and random errors in the ice velocity based on the reference D. The D flux of the AIS increases (or decreases) by 15.1 Gt yr$^{-1}$ with a velocity increase (or decrease) of 1 m yr$^{-1}$ (Fig. 5a). A random error in the ice velocity of $\pm 20$ m yr$^{-1}$ results in a change of $41.1 \pm 0.9$ Gt yr$^{-1}$ in the D for the AIS, which is an order of magnitude smaller than that caused by the systematic error of the same magnitude (Fig. 5b). The change in D for the EAIS due to a random error of $\pm 20$ m yr$^{-1}$ is the largest ($17.4 \pm 0.5$ Gt yr$^{-1}$), followed by the changes observedfor the WAIS ($10.1 \pm 0.5$ Gt yr$^{-1}$), the islands ($9.7 \pm 0.3$ Gt yr$^{-1}$), and the APIS ($4.0 \pm 0.2$ Gt yr$^{-1}$).

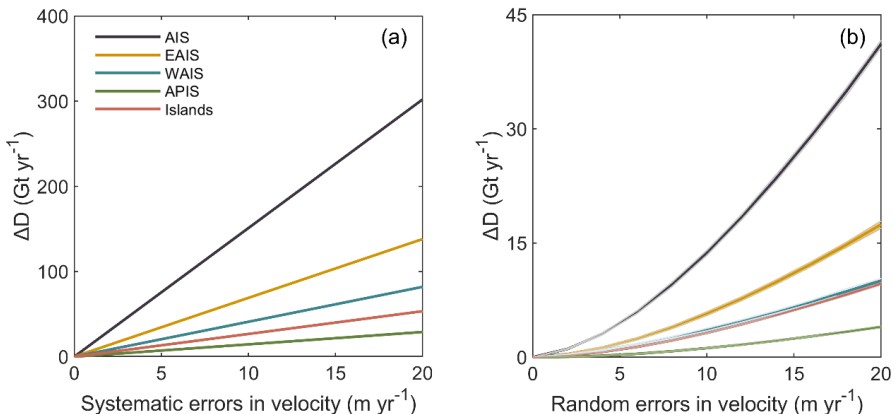

**Figure 5.** The D response to the systematic and random errors in velocity. The colored lines denote (a) the changes in D when the ice velocity at the grounding line increases or decreases by 0–20 m, and (b) the changes in D caused by random errors in the velocity ranging from 0 to 20 m. The lines are the average △Ds of the 100x estimates. The ranges of the 100x estimates are shaded.

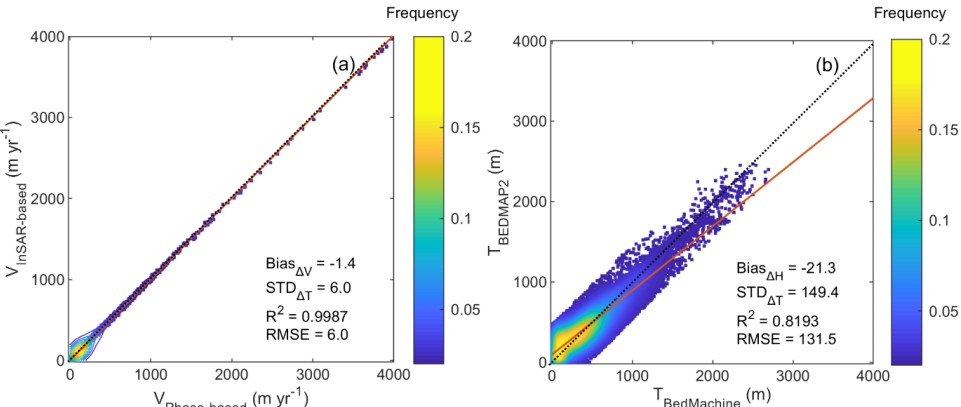

**Figure 6.** Scatter plots of (a) the $V_{InSAR-based}$ versus the $V_{Phase-based}$, and (b) the $T_{BEDMAP2}$ versus the $T_{BedMachine}$, after removing the gross errors greater than three times the standard deviation.

Using almost the same data sources, the error of the MEaSUREs Phase-based velocity is up to 20 cm yr$^{-1}$ over 80% of the Antarctic (Mouginot et al., 2019), while that of the MEaSUREs InSAR-based velocity is 1~17 m yr$^{-1}$ (Rignot et al., 2011a; Mouginot et al., 2012). There is an order of magnitude difference between the errors in the two velocity datasets, and thus, their difference primarily contributes to the error in the InSAR-based velocity data. The D ($V_{InSAR-Based}$) for the AIS is 17.1 Gt yr$^{-1}$ larger than the D ($V_{Phase-Based}$). Basin C-D in the WAIS has the largest △D (2.0 Gt yr$^{-1}$) (Fig. 7). After removing the gross errors greater than three times the standard deviation, the velocity difference between the two is $-1.4\pm6.0$ m yr$^{-1}$ (Fig. 6a),



which corresponds to a difference in D ($\Delta$D) of 21.1 Gt yr$^{-1}$ resulting from a system error of 1.4 m yr$^{-1}$ and a $\Delta$D of 5.9 Gt

yr$^{-1}$ resulting from a random error of 6.0 m yr$^{-1}$ based on the above estimation. This indicates that the contribution of $\Delta$D

is dominated by the system error in the ice velocity. The spatial distribution of $\Delta$D in each basin and flux gate (Fig. 7) confirms

that there is a spatially consistent systematic error in the velocity. Fig. 2b shows that the interannual variation in the D for the

AIS (21.6 Gt yr$^{-1}$) estimated at the pixel scale from 2005 to 2016 is more than three times the uncertainty due to the precision

(i.e., the random error) of the InSAR-based velocity, which suggests that the interannual variation in D due to the change in

the ice velocity cannot be ignored.

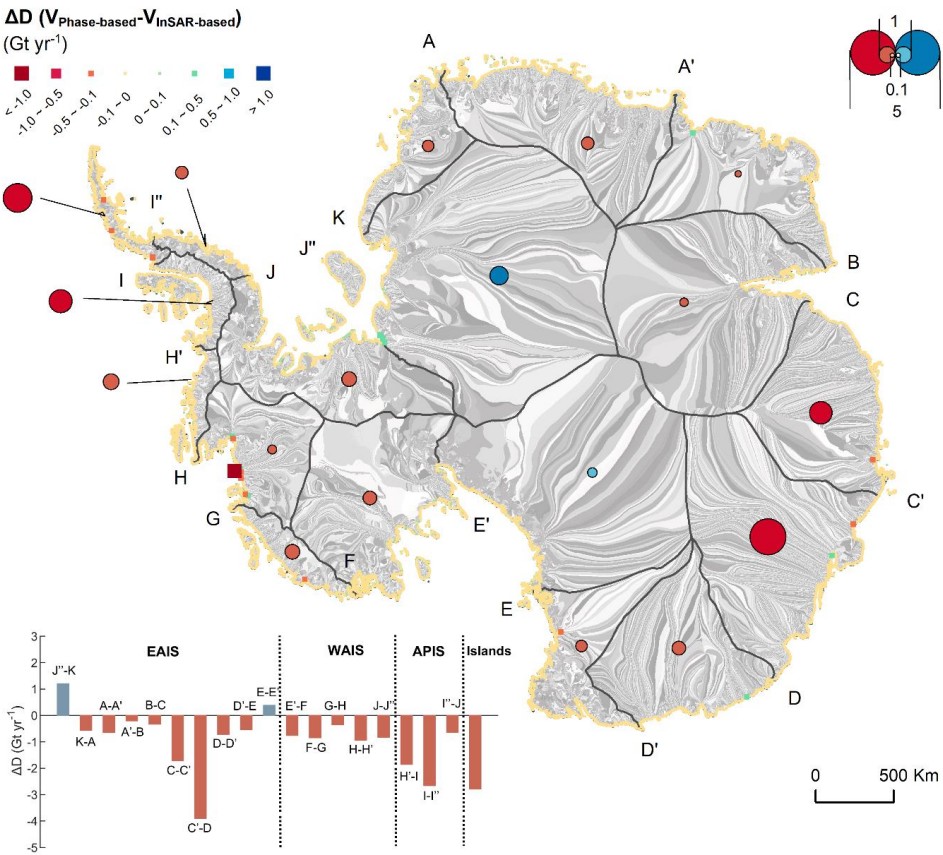

**Figure 7.** The $\Delta$D between the D (V$_{\text{Phase-based}}$) and the D (V$_{\text{InSAR-based}}$) for 18 subregions A–K (black lines), which are color-

coded from red (negative) to blue (positive), with the circle radii proportional to the absolute differences. The bar charts

260        below show the exact differences. The background map shows the flow units flowing into each flux unit.

### 3.2.3 Discharge response to thickness uncertainty

Fig. 8 shows the uncertainty in D due to the systematic and random errors in the ice thickness based on the reference D. Every increase (or decrease) of 1 m in the ice thickness of each flux gate increases (or decreases) the D for the AIS by 3.7 Gt yr$^{-1}$ (Fig. 8a). A random error in the ice thickness of $\pm100$ m only causes a change of $10.5\pm2.7$ Gt yr$^{-1}$ in the Antarctic D and

a change of less than 5 Gt yr$^{-1}$ in the four regions (Fig. 8b). It is surprising that the islands have the largest $\triangle$D of $4.8\pm0.6$ Gt yr$^{-1}$ and the WAIS has the smallest ($1.7\pm0.5$ Gt yr$^{-1}$). For the same magnitude, the $\triangle$D due to the systematic error is an order of magnitude greater than that due to the random error.

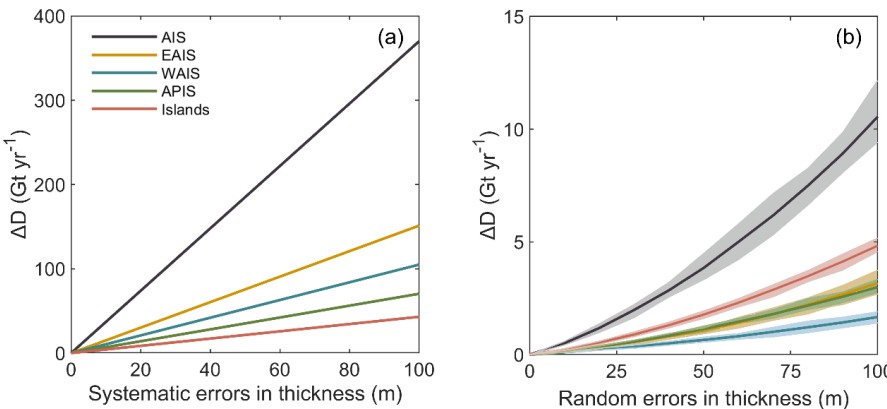

**Figure 8.** The response of D to the systematic and random errors in thickness. The colored lines denote (a) the changes in D

when the ice thickness at the grounding line increases or decreases by 0–100 m and (b) the changes in D caused by random errors in thicknesses of 0–100 m. The lines are the average $\triangle$Ds of the 100x estimates. The ranges of the 100x estimates are shaded.

The D (T$_{BedMachine}$) for the AIS is 17.1 Gt yr$^{-1}$ less than the D (T$_{BEDMAP2}$). The $\triangle$Ds differ for the WAIS (32.9 Gt yr$^{-1}$), APIS (25.7 Gt yr$^{-1}$), islands (24.6 Gt yr$^{-1}$), and EAIS (18.4 Gt yr$^{-1}$). The largest $\triangle$D is for basin C-D in the EAIS (41.5 Gt

yr$^{-1}$), followed by basin H-H' (22.9 Gt yr$^{-1}$), and basin G-H (18.9 Gt yr$^{-1}$) in the WAIS (Fig. 9). After removing the differences larger than three times the standard deviation, the thickness difference between the two datasets is $-21.3\pm149.4$ m (Fig. 6b), which indicates a $\triangle$D of 78.8 Gt yr$^{-1}$ resulting from the systematic error of 23.9 m and a $\triangle$D of 15.8 Gt yr$^{-1}$ resulting from the random error of 149.4 m. This estimated value is smaller than the $\triangle$D (101.7 Gt yr$^{-1}$) between the D (T$_{BedMachine}$) and D (T$_{BEDMAP2}$) for the AIS. The basins and flux gates for which D (T$_{BedMachine}$) is less than D (T$_{BEDMAP2}$) are mostly located on the

dominant ice-shelf thinning regions (Paolo et al., 2015), including the Amundsen Sea Bay (ASE), the Bellingshausen Sea (BS), Wilkes Land, and the West coast of the APIS (Fig. 9). The flux gates for which D (T$_{BedMachine}$) is larger than D (T$_{BEDMAP2}$) are mainly located on significantly thickening large ice shelves, for example, the Ross Ice Shelf, the Ronne and Filchner Ice Shelf, and the Amery Ice Shelf. Therefore, the difference between BEDMAP2 and BedMachine is likely to include the actual thickness change.

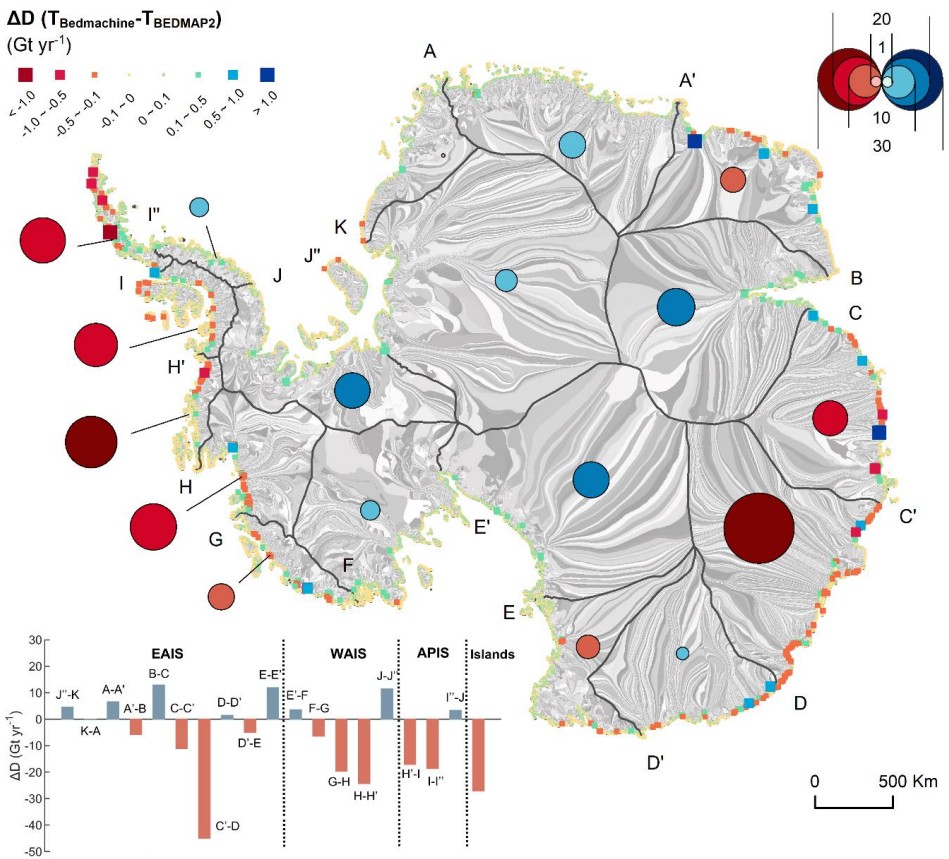

**Figure 9.** The $\Delta$D between the D ($T_{BedMachine}$) and the D ($T_{BEDMAP2}$) for 18 basins A-K (black lines), which are color-coded from red (negative) to blue (positive), with the circle radii proportional to the absolute differences. The bar charts below show the exact differences. The background map shows the flow units flowing into each flux unit.

### 3.2.4 Discharge response to grounding line uncertainty

Compared with the D ($GL_0$) for the AIS, the D ($GL_{Advance}$) and the D ($GL_{Retreat}$) decreased by 278.1 Gt yr$^{-1}$ and 79.6 Gt yr$^{-1}$ for a 1-km advance and a 1-km retreat of the grounding line, respectively. Assuming a linearly distributed uncertainty, this indicates that the D uncertainty due to the precision of the 100-m grounding line is 8.0–27.8 Gt yr$^{-1}$, which cannot be ignored. The uncertainty due to the mis-downstream of the grounding line is several times greater than that due to the mis-upstream of the grounding line (Fig. 10).

Both the advance and retreat thicknesses (or velocities) generally exhibit systematic decreases (Fig. 10), but the estimated $\Delta$D due to a systematic decrease in the ice velocity is more than 10 times greater than that due to a systematic decrease in the thickness. The spatial distribution of the $\Delta$D due to an advance or retreat in the 99 sub-basins and 18 basins generally shows





systematic decreases (Fig. 11), but the $\Delta$D for the advance is much greater than the $\Delta$D for the retreat. This result is consistent with the fact that there is significant basal melt of the ice-shelf near the grounding line (Rignot et al., 2013).

Furthermore, the advance velocities of less than 100 m yr$^{-1}$ distinctly increase (Fig. 10b) and are consistent with the situation in which the advanced grounding line downstream moves from the slow ice flow to the fast ice flow. These results indirectly verify the accuracy of the MEaSUREs grounding line.

The difference between the D (GL$_0$) and the D (GL$_{Advance}$) is mostly attributed to basin H'-I (33.7 Gt yr$^{-1}$) in the APIS, followed by basin F-G (30.7 Gt yr$^{-1}$), and basin G-H (29.6 Gt yr$^{-1}$) in the WAIS (Fig. 11), which is consistent with the

significant ice-shelf thinning in these regions. The D (GL$_{Retreat}$) values of the 18 basins in the EAIS, WAIS and APIS are very close to the D (GL$_0$) values, with a total $\Delta$D of 17.1 Gt yr$^{-1}$, but the D (GL$_{Retreat}$) differs from the D (GL$_0$) by up to 62.5 Gt yr$^{-1}$ in the islands (Fig. 11). The D (GL$_{Retreat}$) of 66.9 Gt yr$^{-1}$ is closer to the D of 77.0 Gt yr$^{-1}$ estimated by Rignot et al. (2019) and the average SMBs in the islands than the D (GL$_0$), which indicates that the real grounding line of the island is likely to be upstream of the MEaSUREs grounding line.

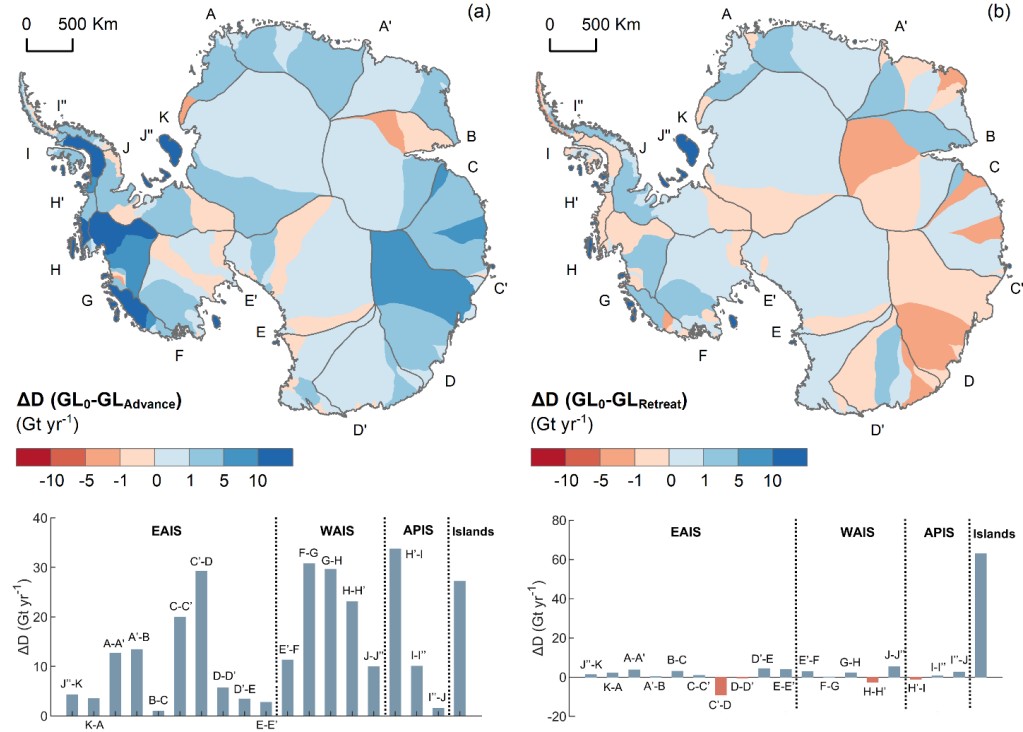


**Figure 10.** (a) Difference between D (GL$_0$) and D (GL$_{Advance}$). (b) Difference between D (GL$_0$) and D (GL$_{Retreat}$) for the 18 sub-regions A–K (black lines), which are color-coded from red (negative) to blue (positive). The bar charts below show the exact differences.

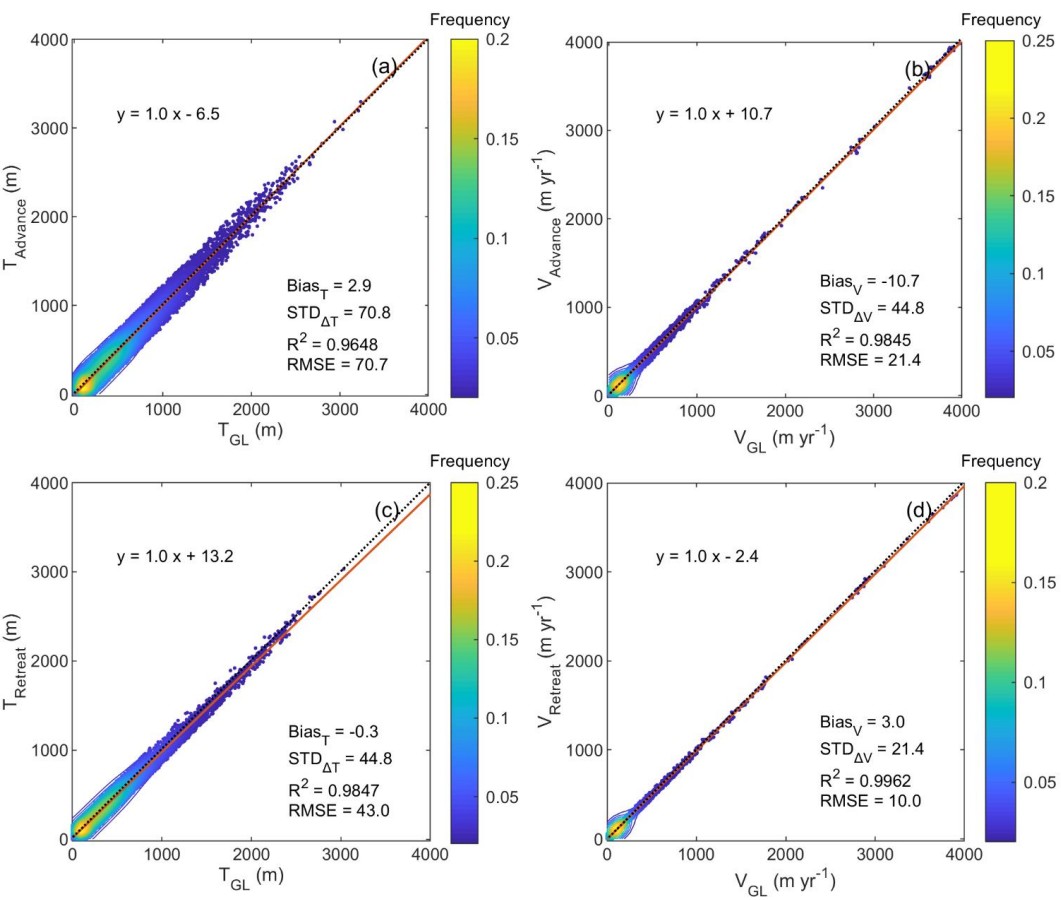

**Figure 11.** Scatter plots of (a) $T_{GL0}$ versus $T_{Advance}$, (b)$V_{GL0}$ versus $V_{Advance}$, (c) $T_{GL0}$ versus $T_{Retreat}$, and (d) $V_{GL0}$ versus $V_{Retreat}$, after removing the gross errors greater than three times the standard deviation.

## 4 Discussion

How much will the global and regional sea level rise over the next decade and beyond? What will be the role of the ice sheets and ocean heat storage? To answer these scientific questions, scientists are required to determine the changes in the total ice sheet MB within 15 Gt yr$^{-1}$ over the course of a decade, as well as the changes in the SMB and D with the same accuracy over all of the ice sheets (Board et al., 2019).

Effectively quantifying the changes in the MB estimation at a fine spatiotemporal scale is the basis for presenting the spatiotemporal complexities of the ice sheet's evolution and reducing the uncertainties in the projection of future changes. In this study, we estimated the ice discharge using the refined IOM, which enables a detailed discharge estimation at the pixel scale, and we compared the results with those obtained using scaling factors. Previous studies using the IOM for Antarctic MB estimation usually simplified or overlooked the accurate estimation of D in the low-velocity regions (Gardner et al., 2018;



Shen et al., 2018; Rignot et al., 2019). For example, Rignot et al. (2019) used a scaling factor to estimate the interannual change in D in the low-velocity regions under the assumption that the ice discharges of low-velocity regions and high-velocity regions have a consistent response to climate change. Our estimations of D for the full velocity range demonstrate that the interannual

variation characteristics of the regions with different velocities are not consistent (Fig. 2), which may indicate their different responses to climate change. The estimation of D using a scaling factor has larger interannual variability than that of the D obtained using the full velocity range, that is, the refined IOM can reduce the uncertainty of the estimation. In addition, the refined IOM can facilitate comparisons among different studies, which enables the identification of which of the three parameters is the dominant contributor to the difference in D in any regional unit. Based on this, we assessed the

methodological and data uncertainties in the MB estimation of the AIS using the IOM. Our results provide an important basis for future MB estimations with the goal of uncovering how the uncertainties can be reduced. However, the interannual variability in the D of high-velocity regions (V > 100 m yr$^{-1}$) is more than six times larger than that in low-velocity regions (V < 100 m yr$^{-1}$). Using the scaling factor to estimate the D in high-velocity regions leads to a small variability difference in D of 3.6 Gt yr$^{-1}$ for the entire AIS, which is relatively small compared to the required uncertainty of 15 Gt yr$^{-1}$, that is, the scaling

factor strategy is acceptable when there are data gaps in the low-velocity regions.

In previous studies, especially for decadal MB estimations, the annual average or linearly fitted SMB was used by default instead of the annual SMB for the annual MB estimation (Gardner et al., 2018; Shen et al., 2018; Rignot et al., 2019). The annual SMB data are probably closer to the real values, and the future change is dominated by short-term, regional snowfall events (Ligtenberg et al., 2013). For example, during the observation period 2003–2020, over 50% of the mass gained in the

EAIS occurred in only two extreme snowfall episodes between 2009 and 2012 (Wang et al., 2021). However, for the decadal MB estimation, the annual SMB affected by the short-term variability could obscure the long-term, secular trends in the SMB change. It is critical to constrain the variability of the observed rate of annual surface mass change for projecting the contribution of the ice sheet to sea-level rise (Rignot et al., 2011c; Wang et al., 2021).

The uncertainty in the SMB, i.e., the input component, is one of the dominant sources of uncertainty in the IOM. Different

techniques are used to measure the SMB, including in-situ observations, satellite observations, reanalysis data, and modeling simulations. SMBs produced by regional climate models (RCMs) are commonly used in the MB estimation of the AIS. These RCMs include RACMO2.3, MAR, and SEMIC, and their related reanalysis datasets include JRA-55, MERRA, and ERA-Interim (Wang et al., 2016; Fettweis et al., 2013; Krapp et al., 2017). The uncertainty of the modeled SMB depends on the model's resolution, the physical parameterization, and the forcing fields (Agosta et al., 2019). Among the RCMs, the

RACMO2.3 is the most widely used SMB data source for the IOM in recent years.

Different spatial resolutions can lead to regional differences in RACMO2.3 SMB results (Shepherd et al., 2018). The comparison of the SMBs from RACMO2.3 indicates that the Antarctic SMB varies by up to 216.7 Gt yr$^{-1}$ for the same model version but for different spatial resolutions (Fig. 4a, Table 4), with the greatest discrepancies occurring on the APIS, particularly



in West Graham Land. The 5.5-km SMB probably better reflects the actual situation of the sharp climate gradients and steep
mountainous topography of the Antarctic Peninsula (Lenaerts et al., 2014; van Wessem et al., 2016), especially the narrow and
elongated shape of West Graham Land (Fig. 4b), which is more sensitive to scaling effects (Quattrochi and Goodchild, 1997;
Marceau and Hay, 1999). Currently, the SMB of the RACMO2.3p2 (5.5 km) only covers the Antarctic Peninsula region. In
some coastal regions, which are smaller in size and are subject to intense climate changes, high spatial resolution SMB products
are expected to reduce the uncertainty in the MB.

The model's physical parameterization also significantly influenced the SMBs. Compared to the RACMO2.3p1, the
RACMO2.3p2 has been updated to include modified snow properties and tuned cloud scheme parameters to increase
precipitation towards the interior of the AIS (van Wessem et al., 2014; van Wessem et al., 2018). The RACMO2.3p1 SMBs of
the basins in the EAIS are systematically smaller than the RACMO2.3p2 SMBs (Fig. 4a). It is likely that the RACMO2.3p1
systematically underestimates the snowfall, and hence, the SMB over the East Antarctic plateau. However, this scheme is
controversial. Rignot et al. (2019) suggested that the RACMO2.3p1 SMB of the EAIS is in better agreement with the in-situ
observations, but their MB estimation indicates that the EAIS has experienced long-term mass loss, which is diametrically
opposite to the MB results obtained using the altimetry, gravity, and reconciled methods (Bamber et al., 2018; Shepherd et al.,
2018; Schroder et al., 2019; Wang et al., 2020).

The uncertainty in D, i.e., the output component, is jointly determined by the uncertainties of the ice velocity, ice thickness,
and the grounding line. The ice velocity, ice thickness, and the grounding line datasets, cover all of Antarctica and are typically
based on remote-sensing observations. The ice velocity measurement technique is relatively mature, and annual ice velocity
measurement data are available. Using almost the same data sources, the precision of the MEaSUREs Phase-based velocity is
up to 20 cm yr$^{-1}$ (Mouginot et al., 2019), while the MEaSUREs InSAR-based velocity has a precision of $1-17$ m yr$^{-1}$ (Rignot
et al., 2011a; Mouginot et al., 2012). Because the uncertainty of D resulting either from the random error or the system error
of the Phase-based velocity of 20 cm yr$^{-1}$ is negligible, the difference in D ($\Delta$D) between them is primarily attributed to the
error in the InSAR-based velocity data. The velocity difference of $-1.4\pm6.0$ m yr$^{-1}$ between the two suggests that the InSAR-
based velocity at the grounding line is overestimated, which gives a $\Delta$D of 21.1 Gt yr$^{-1}$ resulting from the system error and a
$\Delta$D of 5.9 Gt yr$^{-1}$ resulting from the random error, the total of which (21.9 Gt yr$^{-1}$) is close to our result ($\Delta$D of 17.1 Gt yr$^{-1}$)
(Fig. 6b). Excluding the system error, the InSAR-based velocity product (Mouginot et al., 2017; Shen et al., 2018) also meets
the uncertainty requirement (within 15 Gt yr$^{-1}$). Significant changes in the ice velocity were found in previous studies for
certain basins with high mass losses, such as the Thwaites Glacier, where the ice flow rate accelerated by 33% and the D
increased by 75% between 1973–1996 and 2006–2013, respectively (Mouginot et al., 2014). For an unchanged grounding line
position and the same ice thickness product, the annual variation in D is 21.6 Gt yr$^{-1}$, which is about three times greater than
the uncertainty in D due to the random error of the InSAR-based velocity data. Thus, determining the annual variation in the
ice velocity observations is necessary.



The ice thickness is another dominant source of the uncertainty in the D estimation (Morlighem et al., 2020), due to the data gaps in the direct ice thickness measurements obtained using the RES method (Morlighem et al., 2017). The RES thickness data cover only 19% of the grounding line (Gardner et al., 2018). The thickness of the ice shelves is usually filled with indirect ice thickness estimates calculated from the surface height data assuming hydrostatic equilibrium of the ice, such as the

BEDMAP2 thickness. Furthermore, the BedMachine thickness combines the radar-derived thickness with the ice motion vectors, the SMB from RACMO2.3, and the ice elevation (Morlighem et al., 2020). The ice thickness measurement errors of BEDMAP2 and BedMachine range from 30 to 1000 m, and the average error at the grounding line is about 100 m (Rignot et al., 2011b; Rignot et al., 2014). However, our estimated $\Delta D$ due to the difference between the BEDMAP2 and BedMachine thickness is more than 10 times greater than the $\Delta D$ resulting from the random thickness error of $\pm 100$ m. It is impossible to

determine if the total mean and standard deviation of the thickness difference of the $\Delta D$ for the AIS is caused by the systematic error, the random error, or the combined error, which suggests regional distribution differences in the thickness. In addition, the spatial distribution of the $\Delta D$s in the different basins is consistent with the recent changes in thickness of their connected ice shelves (Paolo et al., 2015), which suggests that the $\Delta D$ probably reflects the change in the ice thickness over time.

There is a great deal of controversy regarding the selection of thickness data for the previous MB estimations using the

IOM. The choice of thickness data is not limited to the BEDMAP2 and BedMachine, and there are more approaches to obtaining the ice thickness. For example, Gardner et al. (2018) back-tracked to the RES profiles upstream of the grounding line to acquire more reasonable ice thickness data. Shen et al. (2018) combined the BEDMAP2 and the RES thickness data and re-calculated the ice shelf thickness using altimetry measurements. Rignot et al. (2019) even used five types of thickness data, including the BEDMAP2, the BedMachine, the ice shelf thickness based on ERS‑1 altimetry data, the ice shelf thickness

obtained from a TanDEM-X (TDX) DEM, and the balance thickness. The balance thickness is derived based on the assumption that the ice discharge is equal to the reference multi-year average SMB. Our reference D of 1936 Gt yr$^{-1}$ obtained using the BedMachine thickness is much smaller than the average D of 2217 Gt yr$^{-1}$ for the period 1979–2017 estimated by Rignot et al. (2019), which is mainly caused by the difference between the BedMachine thickness and the balance thickness in the EAIS and APIS. The total flux of 40% in the EAIS estimated using the balance thickness of Rignot et al. (2019) probably contributes

to the significant difference between the MB in the EAIS estimated by Rignot et al. (2019) and those estimated using the altimetry, gravity, and reconciled methods (Bamber et al., 2018; Shepherd et al., 2018; Schroder et al., 2019; Wang et al., 2020).

Considering that the $\Delta D$ due to the random error in the thickness of $\pm 100$ m is only 10.5 Gt yr$^{-1}$, filling the gaps in the RES thickness at the grounding line is the fundamental solution to constraining the uncertainty. Moreover, the $\Delta D$ due to the systematic errors in the ice thickness (or the change in ice thickness) is an order of magnitude greater than that due to the

random errors. In consideration of the observed thickness change near the grounding line in the WAIS and APIS (Smith et al., 2020), it is necessary to observe the annual or decadal thickness change to reduce the uncertainties in the MB estimation.

The MEaSUREs grounding line product that we used was derived from Interferometric Synthetic Aperture Radar (InSAR) data (Rignot et al., 2013), which is considered to be the most precise method with a precision of 100 m (Rignot et al., 2011b; Rignot et al., 2013; Rignot et al., 2014). Our estimation of the $\Delta$Ds corresponding to the changes in the grounding line position

with this expectation suggests that the MEaSUREs grounding line on the AIS, except for the islands, is reasonable. The Antarctic grounding line is retreating, but the 1-km retreat is equivalent to the total retreat for 40 years at the current rate of change of about 25 m (Konrad et al., 2018). According to our evaluation, the 1-km retreat of the current grounding line will result in a decrease in D of 19.9 Gt yr$^{-1}$ on the AIS, except for the islands. Assuming that the change is linear, the annual change in the grounding line for 40 years results in a change in D in these regions of only 0.5 Gt yr$^{-1}$, which demonstrates that the

grounding line can be updated every decade or several decades compared to the other two parameters. However, the 1-km retreat in the grounding line results in a D reduction of 62.9 Gt yr$^{-1}$ in the islands, i.e., about 50% of the original D, which changes more significantly than the total $\Delta$D in the other regions of the AIS. Compared to the original D, the D of a 1-km retreat in the islands is closer to the balanced ice flux (equal to the multi-year average SMB) of 77.0 Gt yr$^{-1}$ (Rignot et al., 2019). As the islands are typically surrounded by fast ice flows, the D of a 1-km advance of the grounding line in the islands

may be significantly greater than the original D in the islands. Therefore, we believe that the true grounding line in the islands is closer to the 1-km retreat MEaSUREs grounding line. The grounding line for multiple islands from Lei et al. (2017) based on DInSAR is similarly backward compared to the MEaSUREs product, which supports this view.

**5 Conclusions**

In this study, we introduced a framework for the uncertainty analysis of the MB estimation using the IOM and assessed

the method and data uncertainties in the Antarctic MB estimation. In the assessment, we present the refined D estimation at the pixel scale. The results demonstrate that it is difficult to use the previous strategies employed in various methods and the recent available data to achieve the goal of the estimation accuracy of the total ice sheet MB change to within 15 Gt yr$^{-1}$, and the total ice sheet SMB and D estimation having the same accuracy. The results of this study are as follows:

(1) The different strategies employed in the methods can contribute to considerable uncertainties in the annual MB

estimation. The annual D estimated at the pixel scale, which demonstrates that the D response to climate change for ice flows with different velocities is not consistent, has a smaller interannual variability than that calculated using a scaling factor in the fastest parts of the glacier compared to the reference D. The strategy of using the yearly averaged SMB instead of the annual SMB is acceptable for decadal MB estimation to constrain variability when the long-term trend is required. However, if the research is related to short-term, regional snowfall events, it is better to use the annual SMB to determine the annual variability.

(2) The uncertainty of the modeled SMB is significantly contributed by the parameterization and scale effect. The uncertainties of RACMO2.3 SMB due to parameterization and the scale effect are up to 20.4 Gt yr$^{-1}$ and 216.7 Gt yr$^{-1}$,

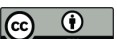

respectively. Different versions of SMB data have considerable differences in the EAIS due to the different physical process constraints adopted. Some areas of rugged terrain such as the APIS, are greatly sensitive to scale effects.

(3) The uncertainty in the future D would most likely be due to the ice thickness data rather than the ice velocity and the grounding line data. The phase-based ice velocity with decimeter-level precision meets the requirements to reach the precision goal of a high precision D estimation, but there is a lack of annual ice velocity products with the same precision. However, the InSAR-based ice velocity with a meter-level precision or ice velocity data with the same precision level, the precisions of which contribute to uncertainties in D of 17.1 Gt yr$^{-1}$, can meet the needs of the annual variation observations because the annual variation in the Antarctic D due to ice velocity changes is much greater than the uncertainty in D caused by the random error in the InSAR-based ice velocity. The InSAR-based grounding line with a 100-m precision can be used to meet the requirements to reach this goal, which results in the D uncertainty being between 8.0–27.8 Gt yr$^{-1}$. However, the InSAR-based grounding line product of the islands is not likely to achieve its nominal precision. The annual variation in D due to the observed rate of change of the grounding line is much smaller than the uncertainty in D, so it can be ignored in the annual estimation. Even ice thickness data with a 100-m precision can be used to meet this goal, which contributes 10.5$\pm$2.7 Gt yr$^{-1}$ to the uncertainty of the D estimation. The ice thickness data gap is the dominant cause of the large uncertainty, which contributes to the D uncertainty of 101.7 Gt yr$^{-1}$ and makes the reference ice thickness for Antarctica controversial. However, this data gap cannot be resolved in the short term. In addition, the annual and decadal variations in the thickness cannot be ignored.

The results of this study increase our understanding of the uncertainty sources and their effects on the estimation of the AIS MB using the IOM, and provide a systematic, comprehensive framework for uncertainty analysis of the MB estimation. In addition, they provide a scientific basis and reference for future improvements in the method and data sets.

**Author Contributions.** Y. Liu, Y. Lin, and Z.Y. conceived, designed, and conducted the experiment. Y. Liu and Y. Lin contributed to the research framework. Y. Lin and Y. Liu performed the data analysis. Y. Liu, Y. Lin, Z.Y., X.C., L.Z., and Q.S. contributed to the analysis of the results. All of the authors contributed to the discussion and writing of the manuscript.

**Competing interests.** The authors declare that they have no conflicts of interest.

**Acknowledgments.** We thank the National Snow and Ice Data Center (NSIDC) for providing the ice velocity, ice-sheet boundary, grounding line, and ice thickness (BedMachine) products. We thank the British Antarctic Survey (BAS) for providing the BEDMAP2, and the Royal Netherlands Meteorological Institute (KNMI) for providing the surface mass balance data. We thank LetPub (www.letpub.com) for its linguistic assistance during the preparation of this manuscript.



**Financial support.** This research was supported by the National Key Research and Development Program of China (Grant
Nos. 2016YFA0600103, 2018YFA0605403, and 2017YFA0603103); National Natural Science Foundation of China (Grant
Nos. 41925027 and 41830536) and Qian Xuesen Lab. - DFH Sat. Co. Joint Research and Development Fund.

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
