# Peer review of "Uncertainties in mass balance estimation of the Antarctic Ice Sheet using the input-output method"

_The Cryosphere, 2021_

## Author Comment (AC1)

**Authors point-to-point response on Referee Comment #1 to tc-2021-325**

**General comments**

*#1*

*I'm a big fan of proper treatment and discussion of uncertainty. I like the idea behind this paper. Unfortunately, I have significant issues with the implementation of the idea. Specifically, the work is incomplete, and difficult to understand. All work could be considered incomplete because any effort can always be done a bit more or better, but by my understanding of uncertainty analysis this work leaves out a lot of sources of uncertainty, and ignores ways that many scientists reduce uncertainty. Or maybe it is there, but suffers from my second issue - it is difficult to understand.*

**Response 1:**

Thanks for the reviewer's comment. We agree and have made efforts to address the issues. Detailed responses are given below.

Most of the mass balance papers rarely mention the uncertainty evaluation in detail, which is a black box for our beginner in mass balance estimation. On the one hand, it is most perplexing that the differences among the mass balance estimates published in authoritative journals are much greater than their respective uncertainties. For example, for the East Antarctic ice sheet (EAIS), in contrast to the mass gain of $58\pm43$ Gt yr$^{-1}$ from 2003–2020 obtained using the gravity method (Wang et al., 2021) and $90\pm21$ Gt yr$^{-1}$ from 2003–2019 obtained using the altimetry method (Smith et al., 2020), mass losses of $82\pm9$ Gt yr$^{-1}$ from 1999–2009 and $51\pm13$ Gt yr$^{-1}$ from 2009–2017 were estimated by Rignot et al. (2019) using the IOM. Rignot et al. (2019) have done a lot work to reduce the uncertainty, but their IOM mass balance of the EAIS is quite different from others. How this could happen? On the other hand, if we are new to do mass balance assessment using IOM method, how can we make others believe our results. This is our original intention of this paper.

We know that the (combined) standard uncertainty is usually used in the IOM uncertainty estimation. We speculate that Rignot et al. (2019) used this method, thus we think it is hard to help us to find the answer. So we tried to get the uncertainty from the random errors generated by Monte Carlo simulation (the Monte Carlo uncertainty) rather than that the (combined) standard uncertainty in the original manuscript. We agree that this point in the manuscript has not been clearly clarified. Though it is really hard to demonstrate all the uncertainty analysis in a paper, we have tried to give more comprehensive uncertainty estimations in the revised version. We have assessed: (1) the (combined) standard uncertainty based on the uncertainty field of the data sets and the given measurement precision, (2) the Monte Carlo uncertainty based on the uncertainty field of the data sets and the given measurement precision, (3) the uncertainty due to system error, (4) the uncertainty presented by the difference between two datasets, between two methods and between two studies.

Moreover, we have restructured the manuscript as suggested by the Referee #2 and have a greater focus on hammering home the key points in the revised manuscript.

The (combined) standard uncertainty is calculated as followings:

The standard uncertainty of D due to the ice thickness ($U\_D_{T_i}$) and ice velocity ($U\_D_{V_i}$) at the pixel scale can be respectively calculated from Eq. (1) and Eq. (2):

$$U\_D_{T_i} = \frac{U_{T_i}}{T_i} \cdot D_i, \tag{1}$$

$$U\_D_{V_i} = \frac{U_{V_i}}{V_i} \cdot D_i, \tag{2}$$

where the $U_{T_i}$ is the thickness uncertainty of flux gate $i$, the $T_i$ is the thickness, the $U_{V_i}$ is the ice velocity uncertainty, the $V_i$ is the velocity, and the $D_i$ is the ice discharge. And for the whole AIS and ice drainage basins, the D uncertainty due to the ice thickness ($U\_D_T$) and ice velocity ($U\_D_V$) can be calculated from Eq. (3) and Eq. (4):

$$U\_D_T = \sum_{i=1}^{n} \frac{U_{T_i}}{T_i} \cdot D_i, \tag{3}$$

$$U\_D_V = \sum_{i=1}^{n} \frac{U_{V_i}}{V_i} \cdot D_i, \tag{4}$$

where the $n$ is the number of flux gates. The D is derived from three components unrelated to and independent of each other. Thus, we used synthetic standard uncertainty to evaluate its accuracy. For the calculations, 917 kg m$^{-3}$ is used for $\rho_{ice}$ and the uncertainties $U_{\rho_{ice}}$ is valued at 5 kg m$^{-3}$ (Griggs et al., 2011). The combined standard uncertainty of D ($U\_D$) is as follows:

$$U\_D = \sqrt{\frac{U_T^2}{T} + \frac{U_V^2}{V} + \frac{U_{\rho_{ice}}^2}{\rho_{ice}}}, \tag{5}$$

To calculate the uncertainty of SMB, we use the relative uncertainty percentage ($u_r$) of the RACMO2.3p2 SMB, which is nearly 4% (Mottram et al., 2021). The standard uncertainty of SMB ($U\_SMB$) is as follows:

$$U\_SMB = u_r \cdot SMB \tag{6}$$

In this way, the uncertainty could be expressed in terms of the magnitude of the value. But this way is unavailable if the uncertainty data sets are not provided, and on the contrast, the second way, comparing results using different dataset combinations, is not limited by the availability of the uncertainty data. The detailed combinations of thickness and velocity products are shown in Table 2 (the manuscript line 170). This comparison may be of certain reference significance to the data set selection. We also compared the Ds of three flux gates (the original grounding-line flux gate, the flux gate for a 1 km advance, and the flux gate for a 1 km retreat).

The Monte Carlo uncertainty evaluates the uncertainty attribution of random errors through generating the random errors based on the precision of the ice velocity and thickness data products we used, which has been described in details in the original manuscript.

**Revised Table 3:**

**Table 3. Data and method uncertainties of the D and SMB (Gt yr$^{-1}$)**

| Components | EAIS | WAIS | APIS | Islands | AIS |
|---|---|---|---|---|---|
| $D_{reference}$ | 895.8 | 750.2 | 157.7 | 132.4 | 1936.0 |
| **U_D** | | | | | |
| **Standard uncertainty** | | | | | |
| U_D($V_{InSAR-based}$) | 33.0 | 13.3 | 3.2 | 10.9 | 60.4 |
| U_D($V_{Phase-based}$) | 8.9 | 5.5 | 1.9 | 3.3 | 19.5 |
| U_D($T_{BEDMAP2}$) | 265.3 | 145.0 | 86.4 | 50.3 | 547.0 |
| U_D($T_{Bedmachine}$) | 93.1 | 50.6 | 30.9 | 33.5 | 208.2 |
| U_D($V_{Phase-based}$ +$T_{Bedmachine}$) | 95.6 | 52.4 | 31.3 | 34.5 | 213.8 |
| **Monte Carlo uncertainty** | | | | | |
| U_D($V_{InSAR-based\_re}$) | 5.3 | 2.2 | 0.3 | 1.8 | 9.6 |
| U_D($V_{Phase-based\_re}$) | 0.9 | 0.6 | 0.2 | 0.3 | 2.0 |
| U_D($V_{re}$ 20 m yr$^{-1}$) | 17.4 | 10.1 | 4.0 | 9.7 | 41.1 |
| U_D($T_{BEDMAP2\_re}$) | 27.9 | 14.5 | 11.6 | 7.3 | 61.4 |
| U_D($T_{Bedmachine\_re}$) | 17.1 | 5.9 | 5.7 | 6.8 | 35.5 |
| U_D($T_{re}$ 100 m) | 3.2 | 1.7 | 3.0 | 4.8 | 10.5 |
| **Uncertainty due to system error** | | | | | |
| U_D($V_{se}$ 20 m yr$^{-1}$) | 138.1 | 80.1 | 29.0 | 53.4 | 302.6 |
| U_D($T_{se}$ 100 m) | 151.2 | 105.1 | 70.4 | 43.1 | 369.8 |
| **Uncertainty presented by difference** | | | | | |
| U_D($V_{InSAR-based}$-$V_{Phase-based}$) | 7.7 | 2.6 | 5.1 | 1.5 | 17.1 |
| U_D($T_{BEDMAP2}$-$T_{BedMachine}$) | 18.4 | 32.9 | 25.7 | 24.6 | 101.7 |
| U_D($GL_{Advance}$-$GL_0$) | -97.2 | -107.1 | -44.4 | 24.7 | -278.1 |
| U_D($GL_{Retreat}$-$GL_0$) | -8.8 | -7.5 | -0.8 | -62.5 | -79.6 |
| U ($D_{ref}$−$D_{Rignot}$) | -214.6 | -24.9 | -172.0 | 55.3 | -356.2 |
| U ($D_{ref}$−$D_{Garder}$) | -56.2 | 11.2 | -62.3 | | |
| U ($D_{ref}$−$D_{Shen}$) | | | | | -171.3 |
| U_MAX(ABS($D_{pixel}$−$D_{scale}$)) | 3.2 | 6.5 | 1.8 | 7.4 | 10.4 |
| **Maximum uncertainty** | | | | | |
| *Max(ABS(U_$D_V$))* | *138.1* | *80.1* | *29.0* | *53.4* | *302.6* |
| *Relative(% of $D_{ref}$)* | *15.4%* | *10.7%* | *18.4%* | *40.3%* | *15.6%* |
| *Max(ABS(U_$D_T$))* | *265.3* | *145.0* | *86.4* | *50.3* | *547.0* |
| *Relative(% of $D_{ref}$)* | *29.6%* | *19.3%* | *54.8%* | *38.0%* | *28.3%* |
| *Max(ABS(U_$D_{GL}$))* | *97.2* | *107.1* | *44.4* | *62.5* | *278.1* |
| *Relative(% of $D_{ref}$)* | *10.9%* | *14.3%* | *28.2%* | *47.2%* | *14.4%* |
| $SMB_{reference}$ | 1153.5 | 649.5 | 253.6 | 61.4 | 2118.0 |
| **U$_{SMB}$** | | | | | |
| **Standard uncertainty** | | | | | |
| U_SMB$_{p2-27}$ (4%) | 46.1 | 26.0 | 10.1 | 2.5 | 84.7 |
| **Uncertainty presented by difference** | | | | | |
| U (SMB$_{p1-35}$-SMB$_{p1-27}$) | -26.1 | -51.6 | -119.4 | -19.6 | -216.7 |
| U (SMB$_{p2-27}$-SMB$_{p2-5.5}$) | | | -51.7 | | |
| U (SMB$_{p1-27}$-SMB$_{p2-27}$) | -78.5 | 3.1 | 39.4 | 15.6 | -20.4 |

| | | | | | |
|---|---|---|---|---|---|
| U (SMB$_{ref}$-SMB$_{Rignot}$) | 78.5 | -3.1 | -39.4 | -15.6 | 20.0 |
| U (SMB$_{ref}$-SMB$_{Gardner}$) | 95.5 | 108.5 | 19.0 | | |
| U (SMB$_{ref}$-SMB$_{Shen}$) | | | | | 217.0 |
| **Maximum uncertainty** | | | | | |
| *Max(ABS(U_SMB))* | *95.5* | *108.5* | *119.4* | *19.6* | *217.0* |
| *Relative(% of SMB)* | *8.3%* | *16.7%* | *47.1%* | *31.9%* | *10.2%* |

**2**

*I think there needs to be a significant effort added to the discussion of random errors vs. systematic bias. Many scientists assume errors are random, and then use this and the mathematical treatment of uncertainty to reduce errors by summing or averaging over many small regions with random error. This may or may not be correct, but should be discussed in detail in this paper. Limited discussion on the distinction between these two types uncertainty, and their treatments in other works, is problematic.*

**Response 2:** We understand the reviewer's point. We used the Monte Carlo method to estimate uncertainties assuming the observation errors are randomly distributed and considered the system error to estimate uncertainties due to the ignorance of the velocity or thickness change. In the revised manuscript, we have discussed it in detail as suggested and added the (combined) standard uncertainties, which have been provided additional data to discuss.

**3**

*The authors make some questionable decisions on use of data to assess uncertainty. For example, they use BedMachine and BedMap, but do not address the uncertainty field provided by BedMachine.*

**Response 3:** We did not write well for the older version  We have added add the D's uncertainty estimations from the uncertainty field of provided by datasets of BedMap, BedMachine, InSAR-based and Phase-based velocities both using the mathematical treatment and Monte Carlo method as revised Table 3.

**4**

*The authors continually refer to an 'uncertainty analysis framework'. What is this? Is 'framework' just a different way of saying 'method'? An "introduction" suggests something new or not seen before. I do not see anything particularly novel in this work. Therefore, if it is not novel, it should at least be comprehensive, but it is not that either.*

**Response 4:** Thank you for your comment. The "uncertainty analysis framework" is a scheme on how to make the uncertainty evaluation of IOM open and transparent. There are more than five research groups to estimate mass balance of Antarctic ice sheet using the IOM. They have done a lot work to reduce the uncertainties. But it is hard to use their method or repeat the work without the details. We presented the details in our uncertainty estimation. For example, to demonstrate the D's details, we showed 58,597 flux gates around the grounding line. We agree that the "framework" expression is inappropriate and have modified it in the revised manuscript.

**5**

*Finally, I feel that the summary of this work could be a simple and clear table showing the*

*uncertainty from different sources in both absolute (Gt/decade?) and relative (% of total uncertainty) terms. Table 3 lists "data uncertainties" but not "method uncertainties" and not relative amounts.*

**Response 5:** Yes. We have added the relative terms and the "method uncertainties" term as the revised Table 3.

**Specific comments**

*The 15 gt/yr target has a reference, but it is not clear that this target is necessary. Furthermore, the reference for that states,*

> *Determine the changes in total ice-sheet mass balance to within 15 Gton/yr over the course of a decade and the changes in surface mass balance and glacier ice discharge with the same accuracy over the entire ice sheets, continuously, for decades to come.*

*And it isn't clear to me how to parse that. Is this 15 Gt/year average over a decade? Or 150 Gt/decade? Or 15 Gt/decade? Something else?*

**Response**: The 15 Gt yr$^{-1}$ is a decadal goal. Scientists are required to determine the changes in the total ice sheet MB within 15 Gt yr$^{-1}$ over the course of a decade, i.e., 15 Gt yr$^{-1}$ per decade or 1.5 Gt yr$^{-2}$, as well as the changes in the SMB and D with the same accuracy over all of the ice sheets (Board et al., 2019). We have deleted the 15 Gt yr$^{-1}$ target to reduce disputes in the revised manuscript.

*Authors use terms without defining them. For example, "scale effect" on L22.*

**Response**: Thank you for your suggestion. We have added the definition of the scale effect in the introduction as suggested in the revised manuscript.

*What does it mean to have an uncertainty on an uncertainty? For example, L24 reports "Uncertainties of [...] X +- Y Gt/yr".*

**Response**: "The uncertainty on an uncertainty" or "$\pm Y$" is the one standard deviation of 100-times uncertainties estimated using a given measurement precision by Monte Carlo method. X is the averaged uncertainty.

*There is often reference to "future D". But everything else is using present or historical data. Why is "D" discussed predictively?*

**Response**: Using the Monte Carlo method, we can estimate the uncertainty of future velocity or thickness with any given precision. But considering the controversy you mentioned, we have deleted such description in the revised manuscript.

*L73 is a motivation for the entire work, "the dominant factors influencing the uncertainties in the MB estimation of the AIS remain unclear." I disagree, the dominant factors are known. Perhaps not their relative scales?*

**Response**: Yes, we have corrected. We have changed "the dominant factors" into "the relative scale of factors" in the revised manuscript.

*L83 "We introduce an uncertainty analysis framework" <-- where?*

**Response**: As is responded in our reply #4, we agree "framework" is not appropriate to express our intention, which has been revised in the revised manuscript.

*L96 "SMB (input)" SMB is output too (runoff, sublimation, evaporation) although runoff is small in Antarctica.*

**Response**: Yes, we have revised this sentence in the revised manuscript.

*L104 D equation also uses ice density.*

**Response**: We have added "ice density" in the sentence as suggested in the revised manuscript.

*L168 Which "original raster" is extended? The velocity raster? thickness? Both?*

**Response**: Both of the velocity and thickness raster are extended. We have clarified this point in this sentence as suggested in the revised manuscript.

*L190-195: This sentence seems out-of-place. It's not about your methods or results of your methods, it is instead about anomalous years. I am unable to see the relevance to the surrounding text or the paper.*

**Response**: We agree and we have removed this sentence in the revised version.

*L447-449: "The strategy of using the yearly averaged SMB instead of the annual SMB is acceptable for decadal MB estimation to constrain variability when the long-term trend is required. However, if the research is related to short-term, regional snowfall events, it is better to use the annual SMB to determine the annual variability." <-- but the motivation for this whole paper is about a 15 Gt/year decade average. In several places there is mention of "annual variation" (see also L467). It isn't clear to me that annual variation matters based on the target "15 Gt/year over the course of a decade". Can these annual variations be considered random and therefore reduce when averaged over a decade?*

**Response**: Thanks for raising this point. The decade uncertainty of SMB can be calculated as follows:

$$U\_SMB_{decade} = \frac{\sqrt{U\_SMB_{yearly}^2 + STD^2}}{9}, \qquad (11)$$

where the $U\_SMB_{yearly}$ is the average uncertainty of SMB during the decade (the relative uncertainty percentage $u_r$ is 4%), and the STD is the standard deviation of the decadal SMB. So we can find the uncertainty of yearly SMB for 2007-2019 is 85.4 Gt yr$^{-1}$, and the uncertainty of decadal SMB for 2007-2019 decrease to 11.9 Gt yr$^{-1}$, which is within the goal of 15 Gt yr$^{-1}$ per decade. We have clarified this point in the revised version.

*L454 - again mention of "future D". Why not also mention "future SMB"?*

**Response**: Using the Monte Carlo method, we can estimate the uncertainty of future velocity or thickness with any given precision. But considering the controversy you mentioned, we have deleted such description of "future D" in the revised manuscript.

*L464 "Even ice thickness data with a 100-m precision" What is 100 m precision? Is that like "thick_100 = round(thick/100)\*100" ?*

**Response**: We appreciate this insightful comment. "100-m precision" is the measurement precision of 100 m, one standard deviation of thickness measurement. Measurement precision is related to random measurement error and is a measure of how close results are to one another. Measurement precision is expressed numerically using measures of imprecision such as one standard deviation calculated from results obtained by carrying out replicate measurements. We have clarified the description in the revised manuscript.

**References**

Board, S. S., National Academies of Sciences, E., and Medicine: Thriving on our changing planet: A decadal strategy for Earth observation from space, The National Academies Press, Washington, USA, 2019.

Gardner, A. S., Moholdt, G., Scambos, T., Fahnstock, M., Ligtenberg, S., van den Broeke, M., and Nilsson, J.: Increased West Antarctic and unchanged East Antarctic ice discharge over the last 7 years, The Cryosphere, 12, 521-547, https://doi.org/10.5194/tc−12-521-2018, 2018.

Mottram, R., Hansen, N., Kittel, C., van Wessem, J. M., Agosta, C., Amory, C., Boberg, F., van de Berg, W. J., Fettweis, X., Gossart, A., van Lipzig, N. P. M., van Meijgaard, E., Orr, A., Phillips, T., Webster, S., Simonsen, S. B., and Souverijns, N.: What is the surface mass balance of Antarctica? An intercomparison of regional climate model estimates, The Cryosphere, 15, 3751–3784, https://doi.org/10.5194/tc-15-3751-2021, 2021.

Rignot, E., Mouginot, J., Scheuchl, B., van den Broeke, M., van Wessem, M. J., and Morlighem, M.: Four decades of Antarctic Ice Sheet mass balance from 1979-2017, Proc. Natl. Acad. Sci. U.S.A., 116, 1095−1103, https://doi.org/10.1073/pnas.1812883116, 2019.

Shen, Q., Wang, H. S., Shum, C. K., Jiang, L. M., Hsu, H. T., and Dong, J. L.: Recent high-resolution Antarctic ice velocity maps reveal increased mass loss in Wilkes Land, East Antarctica, Sci. Rep., 8, 4477, https://doi.org/10.1038/s41598-018-22765-0, 2018.

Smith, B., Fricker, H. A., Gardner, A. S., Medley, B., Nilsson, J., Paolo, F. S., Holschuh, N., Adusumilli, S., Brunt, K., Csatho, B., Harbeck, K., Markus, T., Neumann, T., Siegfried, M. R., and Zwally, H. J.: Pervasive ice sheet mass loss reflects competing ocean and atmosphere processes, Science, 368, 1239−1242, https://doi.org/10.1126/science.aaz5845, 2020.

Wang, L., Davis, J. L., and Howat, I. M.: Complex Patterns of Antarctic Ice Sheet Mass Change Resolved by Time-Dependent Rate Modeling of GRACE and GRACE Follow-On Observations, Geophys. Res. Lett., 48, e2020GL090961, https://doi.org/10.1029/2020GL090961, 2021.

---

## Author Comment (AC2)

**Authors point-to-point response on Referee Comment #2 to tc-2021-325**

**General comments**

*#1 Difficult to follow*
*My main concern with the manuscript is that I found it very difficult to follow and I feel that the manuscript could benefit from a slight restructuring and having a greater focus on hammering home the key points.*
*Small things such as re-structuring the manuscript so it goes through all the uncertainties associated with ice discharge first (e.g. scaling, velocity, ice thickness and grounding line) and then the SMB stuff could make the manuscript easier to follow. At the moment the manuscript is ordered: ice discharge scaling -> SMB -> and then back to other ice discharge stuff, sometimes individual paragraphs contain information on both ice discharge and SMB methods, that makes it difficult to follow.*
*For example, in the methods section you could have new subsections 2.3 Ice Discharge – where you would go through systemically, paragraph by paragraph, what the manuscript focusses regarding ice discharge (e.g. scaling, velocity, ice thickness and grounding line). You could also do the same for a new subsection (2.4) on SMB. You could then structure the results section in a similar manor.*
*The results section should stick to results and not wonder off into speculative discussions*
**Response 1:** We have carefully considered all the comments, and made major revisions. Particularly, we have restructured the manuscript. The revised manuscript has been ordered as: ice discharge -> SMB. The methodological uncertainty of the D has been assessed and discussed in the revised subsections of the ice discharge.

*I think the discussion needs to be more streamlined and more to the point. At the moment it is lengthy and contains a lot of numbers and other bits of text that really don't add anything. I do not think it hammers home the key points. Which brings me to the question what are the key points? Is scaling suitable? What is the dominant source of uncertainty, SMB? Is it imperative that even small grounding line changes are accounted for? Please let the reader know in a concise manor.*
**Response 1:** We agree and have deleted the repeating points and make it more streamlined in the revised manuscript.

*#2 Unclear on Discharge scaling factor*
*You compare discharge using the full pixel values and using a scaling method. I think this is potentially a valuable contribution and I agree there is an unknown impact of using this scaling, particularly going back in time where velocity data is more sparse. But I am not entirely sure how you have done this:*
*Presumably in the innterannual mosaics used for this experiment there are plenty of data gaps, with some regions presumably having no data at some time periods. How do you account for this?*

*How do you get a seemingly ice sheet wide estimate of ice discharge using the pixel value when some of the mosaics will have very large data gaps? More precisely what is a pixel scale estimate, is this simply taking the velocity value directly at the grounding line. A better explanation is needed here.*

**Response 2:** Thank you for your suggestion. There are gaps of annual data. We have used years of data interpolation to fill the gap. Because the ice velocity and ice thickness vary considerably at the pixel scale, we estimated D at a fine scale by discretizing the grounding line into grids of the same cell size (1 km) as the ice thickness and ice velocity data, which divided the MEaSUREs InSAR-based grounding line into 58,597 flux gates at pixel scale. "A pixel scale estimate" is that we calculated D at each flux gate. We have clarified it in the revised manuscript.

*Why did you choose only 100 m, what is the reason for this? For me 100 m yr is still relatively slow flowing ice.*

**Response 2:** We chose 100 m yr$^{-1}$ because more than three quarters of 58,597 flux gates have velocities less than 100 m yr$^{-1}$. The slow flowing ice (less than 100 m yr$^{-1}$) only accounts for 13.3% of the total discharge of Antarctica, which is the possible reason why that the uncertainty due to scaling is small.

*How do you treat the Antarctic Peninsula which has some difficult topography (e.g. narrow fjords) and how does this compare to other studies e.g. (Gardner et al., 2018).?*

**Response 2:** The difficult topography like narrow fjords requires higher spatial resolution, and the 1-km raster we used may omit or simplify these topographies. But we haven't treat the Antarctic Peninsula in the special way. And compared to the ice discharge of AP during 2008–2015 in Gardner et al. (2018) (234±27 Gt yr$^{-1}$), our ice discharge result is much smaller (161.5 Gt yr$^{-1}$). Therefore, our results of the Antarctic Peninsula probably contain errors that were not clarified in the original manuscript. We have added the D's difference between Gardner et al. (2018) s' result and the reference D in the Table 3 and discussed this point in the discussion of the revised manuscript.

**Revised Table 3:**

**Table 3. Data and method uncertainties of the D and SMB (Gt yr$^{-1}$)**

| Components | EAIS | WAIS | APIS | Islands | AIS |
|---|---|---|---|---|---|
| **D$_{reference}$** | 895.8 | 750.2 | 157.7 | 132.4 | 1936.0 |
| **U_D** | | | | | |
| **Standard uncertainty** | | | | | |
| U_D(V$_{InSAR-based}$) | 33.0 | 13.3 | 3.2 | 10.9 | 60.4 |
| U_D(V$_{Phase-based}$) | 8.9 | 5.5 | 1.9 | 3.3 | 19.5 |
| U_D(T$_{BEDMAP2}$) | 265.3 | 145.0 | 86.4 | 50.3 | 547.0 |
| U_D(T$_{Bedmachine}$) | 93.1 | 50.6 | 30.9 | 33.5 | 208.2 |
| U_D(V$_{Phase-based}$ +T$_{Bedmachine}$) | 95.6 | 52.4 | 31.3 | 34.5 | 213.8 |
| **Monte Carlo uncertainty** | | | | | |
| U_D(V$_{InSAR-based\_re}$) | 5.3 | 2.2 | 0.3 | 1.8 | 9.6 |
| U_D(V$_{Phase-based\_re}$) | 0.9 | 0.6 | 0.2 | 0.3 | 2.0 |
| U_D(V$_{re}$ 20 m yr$^{-1}$) | 17.4 | 10.1 | 4.0 | 9.7 | 41.1 |
| U_D(T$_{BEDMAP2\_re}$) | 27.9 | 14.5 | 11.6 | 7.3 | 61.4 |
| U_D(T$_{Bedmachine\_re}$) | 17.1 | 5.9 | 5.7 | 6.8 | 35.5 |

| | | | | | |
|---|---|---|---|---|---|
| U_D($T_{re}$ 100 m) | 3.2 | 1.7 | 3.0 | 4.8 | 10.5 |
| **Uncertainty due to system error** | | | | | |
| U_D($V_{se}$ 20 m yr$^{-1}$) | 138.1 | 80.1 | 29.0 | 53.4 | 302.6 |
| U_D($T_{se}$ 100 m) | 151.2 | 105.1 | 70.4 | 43.1 | 369.8 |
| **Uncertainty presented by difference** | | | | | |
| U_D($V_{InSAR\text{-}based}$-$V_{Phase\text{-}based}$) | 7.7 | 2.6 | 5.1 | 1.5 | 17.1 |
| U_D($T_{BEDMAP2}$-$T_{BedMachine}$) | 18.4 | 32.9 | 25.7 | 24.6 | 101.7 |
| U_D($GL_{Advance}$-$GL_0$) | -97.2 | -107.1 | -44.4 | 24.7 | -278.1 |
| U_D($GL_{Retreat}$-$GL_0$) | -8.8 | -7.5 | -0.8 | -62.5 | -79.6 |
| U ($D_{ref}$−$D_{Rignot}$) | -214.6 | -24.9 | -172.0 | 55.3 | -356.2 |
| U ($D_{ref}$−$D_{Garder}$) | -56.2 | 11.2 | -62.3 | | |
| U ($D_{ref}$−$D_{Shen}$) | | | | | -171.3 |
| U_MAX(ABS($D_{pixel}$−$D_{scale}$)) | 3.2 | 6.5 | 1.8 | 7.4 | 10.4 |
| **Maximum uncertainty** | | | | | |
| *Max(ABS(U_D$_V$))* | *138.1* | *80.1* | *29.0* | *53.4* | *302.6* |
| *Relative(% of $D_{ref}$)* | *15.4%* | *10.7%* | *18.4%* | *40.3%* | *15.6%* |
| *Max(ABS(U_D$_T$))* | *265.3* | *145.0* | *86.4* | *50.3* | *547.0* |
| *Relative(% of $D_{ref}$)* | *29.6%* | *19.3%* | *54.8%* | *38.0%* | *28.3%* |
| *Max(ABS(U_D$_{GL}$))* | *97.2* | *107.1* | *44.4* | *62.5* | *278.1* |
| *Relative(% of $D_{ref}$)* | *10.9%* | *14.3%* | *28.2%* | *47.2%* | *14.4%* |
| **SMB$_{reference}$** | 1153.5 | 649.5 | 253.6 | 61.4 | 2118.0 |
| **U$_{SMB}$** | | | | | |
| **Standard uncertainty** | | | | | |
| U_SMB$_{p2\text{-}27}$ (4%) | 46.1 | 26.0 | 10.1 | 2.5 | 84.7 |
| **Uncertainty presented by difference** | | | | | |
| U (SMB$_{p1\text{-}35}$-SMB$_{p1\text{-}27}$) | -26.1 | -51.6 | -119.4 | -19.6 | -216.7 |
| U (SMB$_{p2\text{-}27}$-SMB$_{p2\text{-}5.5}$) | | | -51.7 | | |
| U (SMB$_{p1\text{-}27}$-SMB$_{p2\text{-}27}$) | -78.5 | 3.1 | 39.4 | 15.6 | -20.4 |
| U (SMB$_{ref}$-SMB$_{Rignot}$) | 78.5 | -3.1 | -39.4 | -15.6 | 20.0 |
| U (SMB$_{ref}$-SMB$_{Gardner}$) | 95.5 | 108.5 | 19.0 | | |
| U (SMB$_{ref}$-SMB$_{Shen}$) | | | | | 217.0 |
| **Maximum uncertainty** | | | | | |
| *Max(ABS(U_SMB))* | *95.5* | *108.5* | *119.4* | *19.6* | *217.0* |
| *Relative(% of SMB)* | *8.3%* | *16.7%* | *47.1%* | *31.9%* | *10.2%* |

*#3 Lack of the highly relevant reference*

*What about other SMB models apart from RACMO? I think some analysis of this has already been done, so it is not necessary to repeat. But I am certainly surprised not to see any mention of the below paper at all in the manuscript, it seems highly relevant.*

*Mottram, R., Hansen, N., Kittel, C., van Wessem, J. M., Agosta, C., Amory, C., Boberg, F., van de Berg, W. J., Fettweis, X., Gossart, A., van Lipzig, N. P. M., van Meijgaard, E., Orr, A., Phillips, T, Webster, S., Simonsen, S. B., and Souverijns, N.: What is the surface mass balance of Antarctica? An intercomparison of regional climate model estimates, The Cryosphere, 15, 3751–3784,*

*https://doi.org/10.5194/tc-15-3751-2021, 2021.*

**Response 3:** Thanks for your suggestion. The RACMO SMB is the most commonly used in the mass balance estimation of Antarctic ice sheet using the IOM. The comparison between RACMO and other SMB have been done in many work just as the reference you suggest. We have added the reference in the revised description of the SMB product and calculated the standard uncertainty of RACMO2.3p2 SMB (revised Table 3) as suggested by Agosta et al. (2019).

*Agosta, C., Amory, C., Kittel, C., Orsi, A., Favier, V., Gallee, H., van den Broeke, M. R., Lenaerts, J. T. M., van Wessem, J. M., van de Berg, W. J., and Fettweis, X.: Estimation of the Antarctic surface mass balance using the regional climate model MAR (1979-2015) and identification of dominant processes, The Cryosphere, 13, 281-296, https://doi.org/10.5194/tc−13-281-2019, 2019.*

*#4 Ice thickness*
*I agree with reviewer 1 that it is odd not include some analysis on the uncertainty product associated with BedMachine.*

**Response 4:** Thanks for the reviewer's comment. In the revised version, we have added the D's uncertainty estimations from the uncertainty field of provided by datasets of BEDMAP2, BedMachine, InSAR-based and Phase-based velocities both using the mathematical treatment and Monte Carlo method.

**Specific comments**

*Line 63-69: It is not clear to me what the 'maximum difference' is here? The difference to what exactly?*

**Response**: The "maximum difference" means that the max diff value among these studies mentioned above in the same available year. We have clarified this point in the revised manuscript.

*Line 67-69: Over what time periods are these comparisons made? Are you directly comparing the average mass balance from the entire rignot record (1979-2017) to the much shorter record of Shen or Gardner (2008-2015), I can not tell? For example, for Pine Island because it is changing so rapidly different estimates of mass balance would be expected over different timescales. So this is not really a fair comparison.*

**Response:** These comparisons are made in the same available year for the studies mentioned above. Rignot et al. (2019) provided the annual results from 1979 to 2017 in supplement materials. Shen et al. (2018) evaluated the AIS mass balance for 2008, 2014 and 2015, and Gardner et al. (2018) calculated mass balance for 2008-2015. We selected the intersections of their time period, including 2008, 2014 and 2015.

*Line 83: I am not sure what you mean by an uncertainty analysis framework*

**Response:** The "uncertainty analysis framework" is a scheme on how to make the uncertainty evaluation of IOM open and transparent. There are more than five research groups to estimate mass balance of Antarctic ice sheet using the IOM. They have done a lot work to reduce the uncertainties.

But it is hard to use their method or repeat the work without the details. We presented the details in our uncertainty estimation. For example, to demonstrate the D's details, we showed 58,597 flux gates around the grounding line. We recognized that the "framework" expression is inappropriate and have modified it in the revised manuscript.

*Line 125: More detail on the scaling factor is needed (See main comments)*
**Response:** Yes. We have added it in the revised manuscript.

*Line 133: New paragraph as you move onto SMB*
**Response:** Done.

*Line 167: What is the source for the original flux gate*
**Response:** The original flux gate is generated based on the InSAR based grounding line of the MEaSUREs Antarctic boundaries (Rignot et al., 2013).

*Line 176-177: Probably a point for the discussion. Also, this is a large number, why is this? I presume it is something to do with the ice thickness used to calculate D – Rignot uses SMB as a base ice flux from the 1970s where there is no suitable ice thickness data?*
**Response:** Thank you for your suggestion. The large number is something to do with the ice thickness. We have discussed in the original version: "Our reference D of 1936 Gt yr$^{-1}$ obtained using the BedMachine thickness is much smaller than the average D of 2217 Gt yr$^{-1}$ for the period 1979-2017 estimated by Rignot et al. (2019), which is mainly caused by the difference between the BedMachine thickness and the balance thickness in the EAIS and APIS."

*Line 188 – discharge, not 'mass balance'*
**Response:** Corrected.

*Line 195: This sentence seems out of place*
**Response:** We have modified it in the revised manuscript.

*Figure 3: Specify in caption on figure that the multi-year average is 12 years*
**Response:** Done.

*Line 255: 'cannot be ignored' – or confirms internal variation in ice discharge driven by climatological factors*
**Response:** Yes. We have revised.

*Line 279 – 285: This is speculative discussion and should not be in the results. Also I would certainly disagree that ice shelves such as the Ross Ronne and Amery are 'significantly thickening', likewise for 'significantly thinning' ice shelves in Wilkes Land – much depends on what timescales you are referring too. I do not think this is a valid conclusion.*
**Response:** Thank you for your comment. We have removed this part in the revised version.

*Line 303: I was not aware that the Measures grounding line covers the whole ice sheet? Are there not gaps? What did you use for the gaps?*

**Response:** We used the MEaSUREs Antarctic boundary dataset (Rignot et al., 2013) for the estimates in different regions and basins because this AIS boundary is consistent with the MEaSUREs InSAR based grounding line.

[Figure]

Figure 2. Shapefile Spatial Coverage Maps (a-e) and GeoTIFF raster map (f). Each file is available separately, the full information content is provided in the file IceBoundaries_Antarctica_v02.shp

*Line 308-310: Again this is discussion. Furthermore, I don't think I understand what you are trying to say in this point*

**Response:** Our point is that the D of a 1-km retreat in the islands is closer to the balanced ice flux (equal to the multi-year average SMB) of 77.0 Gt yr$^{-1}$ (Rignot et al., 2019) compared to the original D. As the islands are typically surrounded by fast ice flows, the D of a 1-km advance of the grounding line in the islands may be significantly greater than the original D in the islands. Therefore, we believe that the true grounding line in the islands is closer to the 1-km retreat MEaSUREs grounding line. The grounding line for multiple islands from Lei et al. (2017) based on DInSAR is similarly backward compared to the MEaSUREs product, which supports this view. We have clarified it more clearly in the revised manuscript.

*Line 318-329; Example of text that is arguably not needed, most of it is repeated elsewhere in the manuscript*

**Response:** Done.

*Line 336-337: I am not sure of the relevance of this sentence. Slow flowing ice only accounts for a small portion of the total discharge of Antarctica, So in absolute terms faster flowing ice is always going to have larger interannual variability.*

**Response:** Thank you for your comment. The slow flowing ice (less than $100\,\mathrm{m\,yr^{-1}}$) only accounts for 13.3% of the total discharge of Antarctica, which is the possible reason why that the uncertainty due to scaling is small. If we choose $200\,\mathrm{m\,yr^{-1}}$ or larger for slow flowing ice, the uncertainty may be larger. We have clarified it in the revised paper.

*Line 343: 'The annual SMB data are probably closer to the real values' – I do not understand what you mean here*

**Response:** The annual SMB data can reflect the climate-related anomalies compared to the multi-year average SMB. For example, in the EAIS, the SMB exhibited two significant positive anomalies in 2009 ($178.0\,\mathrm{Gt\,yr^{-1}}$) and 2011 ($110.8\,\mathrm{Gt\,yr^{-1}}$) compared with the 12-year mean, which were both caused by extreme snowfall episodes.

*Line 318-435: In general this discussion needs to be more streamlined and have less of a focus on repeating points made earlier in the manuscript.*

**Response:** Thank you for your suggestion. We have deleted the repeating points and make it more streamlined in the revised manuscript.

**References**

Agosta, C., Amory, C., Kittel, C., Orsi, A., Favier, V., Gallee, H., van den Broeke, M. R., Lenaerts, J. T. M., van Wessem, J. M., van de Berg, W. J., and Fettweis, X.: Estimation of the Antarctic surface mass balance using the regional climate model MAR (1979-2015) and identification of dominant processes, The Cryosphere, 13, 281-296, https://doi.org/10.5194/tc−13-281-2019, 2019.

Gardner, A. S., Moholdt, G., Scambos, T., Fahnstock, M., Ligtenberg, S., van den Broeke, M., and Nilsson, J.: Increased West Antarctic and unchanged East Antarctic ice discharge over the last 7 years, The Cryosphere, 12, 521-547, https://doi.org/10.5194/tc−12-521-2018, 2018.

Lei, H., Zhou, C., and Chen, Y.: Determination of grounding line on the Amery Ice Shelf using Sentinel−1 radar interferometry data, Adv. Polar Sci., 204-213, https://doi.org/10.13679/j.advps.2017.3.00204, 2017.

Mottram, R., Hansen, N., Kittel, C., van Wessem, J. M., Agosta, C., Amory, C., Boberg, F., van de Berg, W. J., Fettweis, X., Gossart, A., van Lipzig, N. P. M., van Meijgaard, E., Orr, A., Phillips, T., Webster, S., Simonsen, S. B., and Souverijns, N.: What is the surface mass balance of Antarctica? An intercomparison of regional climate model estimates, The Cryosphere, 15, 3751–3784, https://doi.org/10.5194/tc-15-3751-2021, 2021.

Rignot, E., Jacobs, S., Mouginot, J., and Scheuchl, B.: Ice-Shelf melting around Antarctica, Science, 341, 266-270, https://doi.org/10.1126/science.1235798, 2013.

Rignot, E., Mouginot, J., Scheuchl, B., van den Broeke, M., van Wessem, M. J., and Morlighem, M.: Four decades of Antarctic Ice Sheet mass balance from 1979-2017, Proc. Natl. Acad. Sci. U.S.A.,

116, 1095−1103, https://doi.org/10.1073/pnas.1812883116, 2019.

Shen, Q., Wang, H. S., Shum, C. K., Jiang, L. M., Hsu, H. T., and Dong, J. L.: Recent high-resolution Antarctic ice velocity maps reveal increased mass loss in Wilkes Land, East Antarctica, Sci. Rep., 8, 4477, https://doi.org/10.1038/s41598-018-22765-0, 2018.